# Opinion: Recent Developments and Future Directions in Studying the Mesosphere and Lower Thermosphere

John M. C. Plane[1], Jörg Gumbel[2], Konstantinos S. Kalogerakis[3], Daniel R. Marsh[1,4], and Christian von Savigny[5]

[1]School of Chemistry, University of Leeds, Leeds LS2 9JT, UK
[2]Department of Meteorology, Stockholm University, 106 91 Stockholm, Sweden
[3]Center for Geospace Studies, SRI International, Menlo Park, California 94025, USA
[4]Climate and Global Dynamics Laboratory, U.S. National Center for Atmospheric Research, Boulder, Colorado 80307, USA
[5]Environmental Physics, Institute of Physics, University of Greifswald, 17489 Greifswald, Germany

Correspondence to: J. M. C. Plane (email: j.m.c.plane@leeds.ac.uk)

**Abstract.** This Opinion article begins with a review of important advances in the chemistry and related physics of the Mesosphere and Lower Thermosphere (MLT) region of the atmosphere, that have occurred over the past two decades since the founding of *Atmospheric Chemistry and Physics*. The emphasis here is on chemistry, but we also discuss recent findings on atmospheric dynamics and forcings to the extent that these are important for understanding MLT composition and chemistry. Topics that are covered include: observations, including satellite, rocket and ground-based techniques; the variability and connectedness of the MLT on various length- and time-scales; airglow emissions; the cosmic dust input and meteoric metal layers; and noctilucent/ polar mesospheric ice clouds. The paper then concludes with a discussion of important unanswered questions and likely future directions for the field over the next decade.

## 1 Introduction

Figure 1 illustrates the vertical layers of the atmosphere, which are traditionally defined by the temperature profile. The mesosphere extends from the stratopause (~50 km), which is defined by a local temperature maximum caused by stratospheric ozone absorbing solar near-UV radiation, to the mesopause defined by a local temperature minimum around 85 km in summer and 100 km in winter. The thermosphere then starts at the mesopause, and is characterised by a rapid warming with altitude due to the absorption of extreme UV radiation by $O_2$ at wavelengths below 200 nm. Although translational temperatures in the thermosphere can exceed 1000 K during solar storms, at the very low pressures (< $10^{-6}$ bar above 100 km) the vibrational and rotational modes of molecules are typically not in local thermodynamic equilibrium (Brasseur and Solomon, 2005).

The mesosphere and lower thermosphere (MLT) is the region between about 70 and 120 km. Figure 1 shows schematically the important processes governing its composition and chemistry. The MLT is subject to high energy inputs from space, in the form of solar electromagnetic radiation and energetic particles (mostly electrons and protons of solar origin (Sinnhuber et al., 2012)). The resulting photodissociation, photo-ionization and high-energy collisions generate radicals and ions, often with

internal excitation. The dominant process is photodissociation of $O_2$ through absorption in its Schumann-Runge continuum (130 – 175 nm) and the Schumann-Runge bands (175 – 195 nm), with a less important contribution from $O_3$ photolysis

(Mlynczak et al., 2013). This generates atomic O which participates in highly exothermic reactions (Section 4), converting chemical potential energy into kinetic energy. A roughly similar amount of molecular kinetic energy is deposited from below by the breaking of gravity waves. The dominant cooling process is via emission at 15 µm from $CO_2$; its degenerate bending vibrational mode is efficiently excited by collision with O atoms (Castle et al., 2012). This is also the region where cosmic dust particles entering the atmosphere ablate, injecting a range of metals like Fe, Mg and Na (Plane et al., 2015). The MLT is

therefore subject to extremes: pressures falling from 0.1 mbar to below 0.1 µbar (above the turbopause at a pressure around ~0.5 µbar, molecular diffusion becomes more important than eddy diffusion in transporting constituent species); and temperatures ranging from below 100 K (the coldest part of the planet) to over 2500 K in ablating cosmic dust particles.

Much of what we know today about this atmospheric region is the result of ground- and space-based observations (i.e. remote sensing), together with *in situ* rocket-borne measurements. The atmospheric models which are required to understand these

observations quantitatively depend on laboratory investigations of the underlying fundamental processes, complemented by *ab initio* theoretical calculations and reaction rate theories.

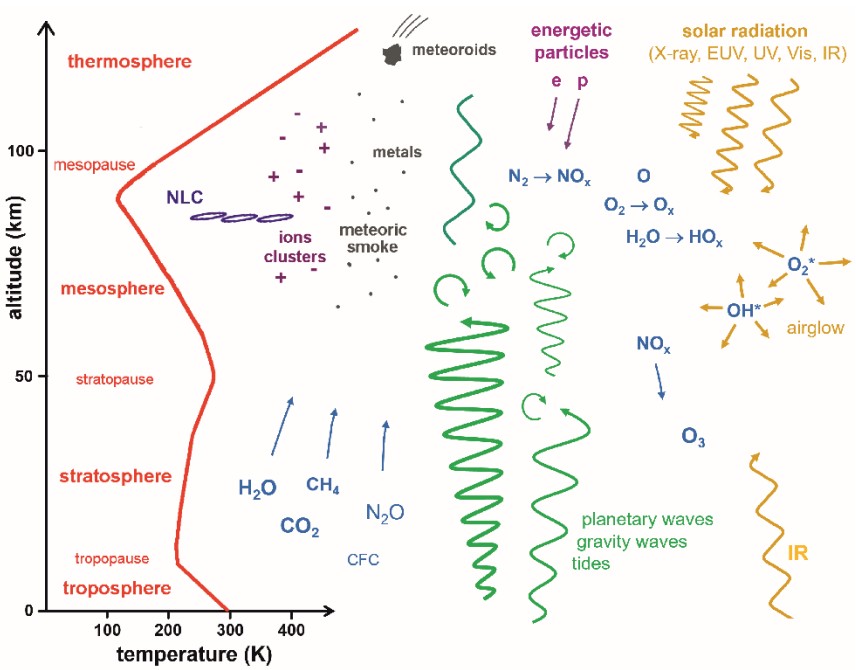

**Figure 1**. Schematic overview of important processes governing the composition and chemistry of the mesosphere and lower thermosphere (MLT). The vertical temperature structure is the basis for defining the "layers" of the atmosphere. The

temperature profile to the left (red line) is representative for high-latitude summer conditions, depicting the extremely cold mesopause that allows for the existence of noctilucent clouds (NLCs). Wave activity is central for connecting the MLT region to the dynamics of the lower atmosphere (shown schematically with green arrows). A large portion of gravity waves break and

deposit their momentum in the MLT region, thus driving the MLT general circulation and controlling thermal structures and transport patterns. Important energy inputs from above are solar radiation, including energetic radiation in the ultraviolet and

X-ray spectrum penetrating to the MLT (brown arrows), and energetic particle precipitation (purple arrows) connected to solar wind and geomagnetic activity. Both drive dissociation and ionization of major constituents, thus initiating a wealth of chemistry that includes excited and ionized species. Through global-scale transport this also affects chemistry at lower altitudes. The upward transport (on average) of $H_2O$, $CH_4$, $CO_2$, $N_2O$ and chlorofluorocarbons (CFC), and downward transport of $NO_x$ ($NO + NO_2$) are shown with blue arrows. A prominent feature of MLT photochemistry is the emission of airglow

(brown arrows arranged in a star). The influx and ablation of meteoroids lead to a complex chemistry involving metal species and ultimately the formation of meteoric smoke particles, which affect charge balance and ice cloud nucleation.

## 2 Observations

### 2.1 Satellite observations

The last two decades have been a golden age in terms of satellite observations of mesospheric composition. Missions with instruments capable of observing the MLT have included Envisat, Odin, AIM, SCISAT, and TIMED (see the acronym list at the end of the paper). The atmospheric density in the MLT is too high for *in situ* measurements by satellites because of the resulting aerodynamic drag, and so satellite instruments rely on remote sensing: observing emissions or the attenuation of solar and stellar irradiance to infer constituent densities. In the first decade of this century there were no fewer than 9 satellite

instruments making observations of mesospheric ozone (Smith et al., 2013). Taken together, a global view of the ozone distribution emerged which revealed that photochemistry dominates variations in the lower mesosphere, but atmospheric dynamics (in particular, the migrating tides) affects ozone at the secondary maximum around 92 km. Model studies revealed that the variations in $O_3$ were predominantly caused by temperature variations changing reaction rates, and by the vertical transport of atomic O from above 95 km. Atomic O, atomic H and OH have been inferred from TIMED-SABER observations

of OH Meinel emissions and ozone emissions at 9.6 µm (Fytterer et al., 2019; Mlynczak et al., 2014; Mlynczak et al., 2018; Panka et al., 2018; Panka et al., 2021; Smith et al., 2010).

Satellite observations of NO have been central for studies of both dynamics and ionization effects in the MLT. Over the past two decades, measurements of NO density in the MLT have been carried by the following spacecraft/spectrometer combinations: MIPAS/ENVISAT (Bermejo-Pantaleón et al., 2011), SCIAMACHY/ENVISAT (Bender et al., 2017),

Odin/SMR(Sheese et al., 2013), ACE/FTS (Boone et al., 2005) and AIM/SOFIE (Hervig et al., 2019a). Observations of NO and excited NO infrared emissions have been particularly useful in detecting the effect of atmospheric tides  on the MLT composition. The large differences in sunrise and sunset HALOE NO concentrations reveal the dominant role of the migrating tide in the vertical transport of NO at low latitudes (Marsh and Roble, 2002; Marsh and Russell, 2000). Analysis of TIMED

wind and temperature measurements showed that longitudinal density variations in NO, measured by the Student Nitric Oxide Explorer

(SNOE), are likely due to the diurnal eastward wavenumber 2 and 3 non-migrating tides (Oberheide and Forbes, 2008). These tides originate in the lower atmosphere and, through their temperature, vertical wind and density perturbations affect NO 5.3-μm emissions in the lower thermosphere (Oberheide et al., 2013). The use of NO to study ionization effects in the MLT is discussed further in section 3.4.

The long duration of satellite observations of the MLT has meant that decadal timescale variations in constituents are now being detected. For example, Odin/SMR has observed CO for over 18 years (Grieco et al., 2020), showing higher concentrations during solar maximum as well as shorter-term variations associated with the semi-annual oscillation and sudden stratospheric warmings. ACE-FTS (on board SCISCAT-1) and SABER observations have been used to detect trends in $CO_2$, with the latest analysis producing values consistent with the anthropogenic trends of about 5% per decade (López-Puertas et al., 2017; Rezac et al., 2018). Earlier analysis had shown larger trends (Emmert et al., 2012) but this is now thought to be sampling biases in the trend analysis (Rezac et al., 2018). The trends in $CO_2$ have almost certainly contributed to a cooling and contraction of the MLT over the last two decades, where contraction is defined as a decrease in the vertical distance between the same pressure levels (Dawkins et al., 2023; Mlynczak et al., 2022).

Lastly, near-global satellite observations of the meteoric metal layers (Section 5) are available: Na and K have been observed since 2004 by the OSIRIS instrument on Odin (Dawkins et al., 2014); Na by SCIAMACHY (Koch et al., 2022) and GOMOS (Fussen et al., 2010) on Envisat; and Mg and $Mg^+$ by SCIAMACHY (Langowski et al., 2015). A very recent study shows good agreement between Na measured by OSIRIS, a ground-based lidar, and the whole atmosphere model WACCM (Yu et al., 2022). Sporadic *E* layers are ionospheric irregularities that consist of high concentrations of metallic ions and electrons in narrow layers (Section 5.2); the electron density can be measured by radio occultation using satellites like COSMIC-1 (Chu et al., 2014), providing a detailed view of sporadic *E* layer morphology (Arras et al., 2022; Xu et al., 2022).

## 2.2 Sounding rockets

*In situ* measurements are needed for many of the scientific topics addressed in this Opinion article. In the mesosphere and lower thermosphere, *in situ* measurements can only be provided by sounding rocket. Satellite-based studies of the MLT are largely restricted to the detection of electromagnetic radiation in various spectral regimes (Section 2.1), and the corresponding remote-sensing techniques have provided us with a wealth of global data on the MLT. However, there are important limitations to these techniques. These include, for example, the inability to measure small-scale structures, analyse complex local interactions, measure charged species, or collect samples. An example of important progress based on sounding rockets is our understanding of the microphysics related to mesospheric particle populations and charging processes (e.g. Friedrich and Rapp (2009); Baumann et al. (2013)). Also, understanding neutral chemical processes often needs detailed local information on the distribution of relevant species and transport processes, not accessible to satellites. A prominent example is airglow (Section 4) and the understanding of the photochemical pathways involved in the generation of various emissions (e.g., Kalogerakis

(2019a); Lednyts'kyy and Savigny (2020); Grygalashvyly et al. (2021)), which then in turn serve as a quantitative basis for remote sensing applications. A key species for many chemical and radiative processes is atomic O as the major carrier of chemical energy in the region. Traditional *in situ* techniques like resonance fluorescence at ultraviolet wavelengths (Dickinson et al., 1980) have been complemented by new developments like electrochemical measurements of atomic oxygen (Eberhart et al., 2015).

The quest to access small-scale structures has recently been extended to three dimensions, leading to new developments beyond the traditional 1-dimensional approach of sounding rocket experiments. New techniques involve the ejection of secondary instrument modules from a primary rocket payload (e.g. Bordogna et al. (2013); Strelnikova et al. (2015)), combined with 2- and 3-dimensional observations using ground-based imaging, radar and/or lidar.

An interesting class of sounding rocket applications is release experiments. These can involve the release of passive tracers for studies of dynamical and transport processes (Gelinas et al., 2006; Larsen, 2002), but can also involve controlled active modification of the local chemical and charging environment (e.g. Collins et al. (2021)). An important quest that has not really been achieved yet is the sampling of material for subsequent analysis on the ground. Efforts have been made to sample Meteoric Smoke Particles (MSPs – Section 5), but so far without conclusive results (Hedin et al., 2014).

## 2.3 Ground-based observations

Passive ground-based measurements applying imaging, spectroscopy, or a combination of both, have long been an important source of information about physical and chemical processes in the MLT. Imaging techniques with sufficient sensitivity and resolution have been developed for dynamics studies covering small-scale gravity waves, instabilities and turbulence. Important observables for such studies have been both nightglow emissions (e.g., an all-sky temperature mapper (Pautet et al., 2014)) and noctilucent clouds (NLCs) which are also known as polar mesospheric clouds (PMCs) (Baumgarten and Fritts, 2014; Kaifler et al., 2023). As for ground-based spectroscopic studies of nightglow chemistry, the past two decades have seen renewed interest particularly in the use of high-resolution spectroscopic data from astronomical telescopes. Slanger and Copeland (2003) and Cosby and Slanger (2007) reviewed early stages of this development. In Section 4 we discuss more recent advances e.g. concerning OH and metal oxide nightglow emissions.

In terms of active ground-based measurements, both lidars and radars have continued to advance our understanding of the MLT. Observations of the metal layers in the MLT (Section 5) have been conducted for more than 60 years, using the techniques of twilight photometry, resonance lidar, rocket-borne mass spectrometry, and optical remote sensing from satellites (Plane, 2003; Plane et al., 2015). Here, we focus on significant lidar developments in the past 10 years. The metal Ni has been observed for the first time, by resonance lidar in the near-UV. Although the first paper (Collins et al., 2015) reported an unexpectedly high density of Ni compared to the well-studied Fe layer, more recent measurements (Gerding et al., 2019; Jiao et al., 2022) show that the Fe/Ni and Na/Ni ratios are close to those expected from modelled injection rates of these three elements (Section 5.1). Another important development is the determination of the vertical fluxes of Na and Fe atoms in the MLT using a Na Doppler lidar and a Fe lidar to measure the correlation between the vertical wind velocity and metal density

(Gardner et al., 2014; Huang et al., 2015). These fluxes are important for constraining the total cosmic dust input into the atmosphere (Section 5.4), as well as using the metal atoms as tracers of vertical transport in this region.

Technological developments in resonance lidars now permit measurements of metal atoms such as Na, Fe and K up to nearly 200 km in the thermosphere (e.g. Raizada et al. (2015); Liu et al. (2016); Tsuda et al. (2015); Chu et al. (2020); Wu et al. (2022)). Besides providing direct temperature profile measurements in this region, these measurements have revealed that the

155 neutral metal atoms often occur in pronounced layers, demonstrating unexpected couplings between the neutral and ionized atmosphere. This has been particularly apparent when simultaneous measurements of two metals are made (e.g. Fe and Na (Chu et al., 2020), Ni and Na (Wu et al., 2022)).

There have also been intriguing recent developments in Rayleigh/Mie/Raman lidar systems. Pushing power and sensitivity, lidar observations of NLCs have now become possible with a temporal resolution down to 1 s, which opens completely new

possibilities for studying small-scale processes in the MLT (Schäfer et al., 2020). Technical improvements on both the transmitter and receiver side have also been the basis for extending Doppler lidar wind measurements all the way up to the MLT (Hildebrand et al., 2017).

Radar observations have made a number of important contributions to MLT science. First, polar mesospheric summer echoes are intense radar backscatter echoes in the VHF and UHF frequency range that appear to be caused by plasma inhomogeneities

which arise from turbulence and are maintained against diffusion by the attachment of charges to ice particles in NLCs (Section 6) (Rapp and Lübken, 2004). Radar observations, which are not constrained by tropospheric weather conditions (unlike lidar), have enabled long-term trend measurements of polar mesospheric summer echoes (PMSE) and hence NLCs (Latteck et al., 2021). A related topic of current research is polar mesosphere winter echoes (PMWE) and their relationship to atmospheric turbulence and possibly MSPs (Strelnikov et al., 2021).

Second, radars can be used to measure the absolute electron density vertical profile; when combined with lidar measurements (e.g., of $Ca^+$ and K (Delgado et al., 2012)), the coupling between the neutral and ionized atmosphere, including space weather effects (Section 3.4), can be studied. Third, observations of meteor head echoes with high performance large aperture radars have provided insights into meteoroid differential ablation (Janches et al., 2009) and fragmentation (Malhotra and Mathews, 2011), as well as the total dust input – though this requires careful analysis of the individual radar's capability to detect small,

slow dust particles (Janches et al., 2017). Fourth, MSPs have been detected by incoherent radar scatter: the relatively heavy charged particles cause a distinctive line shape in the radar spectrum as a result of different diffusion modes in the plasma, allowing size and number density to be retrieved (Strelnikova et al., 2007).

## 3 Variability of the MLT

### 3.1 Thermal balance and energy budget

To understand the chemical state of the MLT, one must first consider how energy is absorbed, redistributed and emitted within this region of the atmosphere (Feofilov and Kutepov, 2012). The primary energy input in the MLT is the absorption of solar

radiation, just as in the stratosphere. However, what distinguishes the MLT is that the solar heating cannot simply be calculated from the divergence of the solar flux, i.e., solar radiation is not thermalized locally. The O and N atoms produced by photodissociation, and indirectly by photoionization, of $O_2$ and $N_2$ undergo exothermic chemical reactions that thermalize the incoming solar energy (unless the energy is lost via airglow), but this may occur at a different time from when, or location from where, the initial photon was absorbed (Mlynczak and Solomon, 1993). One example of this transport of chemical potential energy is the downward diffusion of atomic O created in the thermosphere into the mesosphere where three-body recombination with $O_2$ converts O to $O_3$. Eventually the $O_3$ will be lost via catalytic reactions, mostly with hydrogen or nitrogen-bearing species. Thus, an odd oxygen (O or $O_3$) is converted back to $O_2$ and the energy of a photon absorbed in the thermosphere warms the mesosphere.

Uncertainties in the efficiency by which solar energy is absorbed and redistributed arise from several sources. First, there are uncertainties in the chemical reaction rate coefficients and absorption cross-sections, and knowledge of the background atmospheric state. Second, uncertainty stems from the distribution of the minor constituents (e.g., $O_3$, $H_2O$, NO, O, H and OH) that can act as absorbers or participate in exothermic chemical reactions. The picture of energy transport of heat is further complicated by the fact that the mesosphere is not in local thermal equilibrium, and cooling rates depend not only on temperature but also on the distribution of species such as $CO_2$, O and NO.

Both the background state and the distribution of minor constituents depend on waves that propagate into and can potentially dissipate in the MLT, or that originate in the region. The clearest example is that the mesopause is coldest in summertime, being driven out of radiative balance by adiabatic cooling arising from a mean meridional circulation which is driven by gravity wave momentum deposition (Brasseur and Solomon, 2005). The same circulation pattern brings $H_2O$ and $CO_2$ upwards over the summer pole, and NO and CO downwards from the thermosphere into the wintertime polar mesosphere at the opposite pole (Garcia et al., 2014; Lossow et al., 2009; Smith, 2012). In addition, the wave dissipation itself leads to vertical transport of minor constituents and determines the altitude at which molecular diffusion begins to separate constituents by their mass (Smith et al., 2011). Consequently, much of our understanding of these wave processes has come indirectly from analysis of the distribution of airglow (Section 4), and chemical tracers such as the meteoric metal layers (Section 5). MLT dynamical variability and its connections to the lower atmosphere and to geospace are discussed in the remainder of this Section.

### 3.2 Inter- and intra-hemispheric coupling

One of the most exciting discoveries in the last 20 years is interhemispheric coupling, which describes teleconnection processes mediated by gravity waves, e.g., thermal and dynamical responses at the polar summer mesopause caused by thermal and dynamical conditions in the lower winter stratosphere. The phenomenon was first described in the context of the unprecedented mid-winter major warming in September 2002 (e.g. Becker and Fritts (2006)). The gravity wave-driven processes are essentially instantaneous e.g. the thermal-dynamical signature of a northern hemisphere stratospheric warming can reach the polar summer mesopause region above Antarctica with a time lag of only a few days, as shown by Karlsson et al. (2009b). The basic mechanism, which was described in detail by Karlsson et al. (2009a) and Körnich and Becker (2010), involves the

following processes. The zonal wind in the mid-latitude winter stratosphere affects the filtering of gravity waves and the strength of westward gravity wave drag in the mesopause region of the winter hemisphere, which governs the residual circulation in the MLT. This in turn affects the vertical motion and the adiabatic temperature perturbation in the mid- and high latitude mesosphere. The entire chain of effects works in such a way that the temperature at the polar summer mesopause is positively correlated to the temperature in the lower polar winter stratosphere. The interhemispheric coupling phenomenon is particularly relevant for the variability of NLC/PMCs in the polar summer mesopause and this has been used in several pioneering studies (Gumbel and Karlsson, 2011; Karlsson et al., 2007). The typical interhemispheric coupling pattern in the middle atmospheric temperature field may be triggered by altered planetary wave activity and/or stratospheric warmings, and has also recently been proposed to be present in observational data in the context of a possible Madden-Julian-Oscillation (MJO) signature in the middle atmosphere (Hoffmann et al., 2022). In addition, it has been suggested that the dynamical/thermal effects of volcanic eruptions may affect the interhemispheric coupling pattern (Wallis et al., 2023). It is important to note that the underlying mechanisms are not fully understood for the latter two examples.

Interhemispheric coupling involves intra-hemispheric vertical coupling by gravity waves as outlined above. Intra-hemispheric coupling can also lead to significant variability in the MLT region, particularly at high latitudes. One of the most spectacular manifestations of this coupling is the strong mesospheric cooling during and after stratospheric warming events (e.g. Cho et al. (2004); Pedatella et al. (2014)). The reduction (or even reversal) of the zonal westerly winds in the winter middle atmosphere is thought to reduce (or reverse) the gravity wave drag in the MLT. The altered meridional circulation in the MLT may lead in turn to a reduced mesospheric downwelling above the pole and consequently reduced adiabatic warming. For similar reasons, the onset of the NLC season in the southern hemisphere is directly coupled to the persistence of the Antarctic polar vortex and the time of its breakdown. The earlier the stratospheric wind transition occurs, the earlier the NLC season starts (Gumbel and Karlsson, 2011). A late breakdown of the Antarctic polar vortex may also lead to anomalous mesopause jumps (e.g. Lübken et al. (2017)), corresponding to a sudden increase of the mesopause altitude accompanied by a reduction of mesopause temperature. These jumps require very specific dynamical conditions with continuing eastward flow in the stratosphere and westward flow in the mesosphere.

### 3.3 Solar irradiance effects

The MLT region is significantly affected by solar variability, in several different ways. Periodic or quasi-periodic variations of the electromagnetic radiation emitted by the sun occur at different temporal scales, the most important ones being the 11-year solar cycle (known as Schwabe cycle) and the 27-day rotational cycle. The 11-year cycle is related to the reversal of the Sun's magnetic field and is part of the 22-year Hale cycle. For electromagnetic radiation, these effects are most prominent at short wavelengths, including the vacuum and extreme ultraviolet (VUV, EUV) and soft x-rays. The 27-day cycle is caused by the differential rotation of the Sun, with a period which is slightly variable but averages close to 27 days. In addition to variations in the solar radiation reaching the Earth, the MLT is also affected by solar energetic particles, particularly following

coronal mass ejections (CMEs). In this Section we summarize the current knowledge with a focus on the effects of variable solar electromagnetic radiation.

Solar 11-year and 27-day cycles have been observed in many mesospheric parameters, including the abundance of $O_3$ (e.g.
Hood (1986)), OH (Fytterer et al., 2015; Shapiro et al., 2012), O (Lednyts'kyy et al., 2017), $H_2O$ (e.g. Thomas et al. (2015)) and NO (Hendrickx et al., 2015) as well as temperature (e.g., Hood (1986), Beig et al. (2008), NLCs/PMCs (Robert et al., 2010; Thurairajah et al., 2017) and radio wave reflection heights (von Savigny et al., 2019). Note that some of the solar 27-day effects in atmospheric parameters are caused by variations in solar electromagnetic radiation, while the signatures in other parameters are probably mainly related to variations in the solar wind (e.g. in NO and OH at high latitudes). The relative contributions of variations in the solar wind and variations in solar electromagnetic radiation are not well understood for many of these parameters. Beig (2011) provided a comprehensive review of the 11-year solar cycle effects on temperature in the MLT region, based on experimental data sets. In the southern hemisphere the temperature response to solar variability appears to be larger at higher latitudes, whereas no clear latitude dependence is apparent in the northern hemisphere. The temperature sensitivity (i.e., the temperature change per change in solar proxy) to solar forcing is on the order of $2 - 4$ K/(100 sfu) in the mesopause region. It is remarkable that for several of these parameters – e.g. [O], temperature, NLC albedo and radio reflection heights – the sensitivities to solar forcing for the 11-year cycle and the 27-day cycle agree within uncertainties. This suggests that the same underlying physico-chemical processes drive the atmospheric response to solar variability at these very different time scales. This may not be surprising, but it implies that the relevant mechanisms are relatively rapid, faster than a few weeks. Solar 27-day signatures in middle atmospheric parameters are often relatively easy to extract with high statistical significance if the available data sets are sufficiently long. However, identifying the underlying physico-chemical processes is usually quite complex and many aspects are not well understood. Potential mechanisms for solar variability in chemical species abundances are variable photolysis of the corresponding chemical species or their precursors, temperature effects on chemical equilibria and rate coefficients, dynamically driven changes in species abundances or temperature, and EUV photoionization above 90 km. Solar effects on MLT temperature may be caused by variable solar diabatic heating, by changes in chemical heating associated with solar composition changes (e.g. via $H + O_3$), or by dynamical coupling processes propagating a solar response from the lower atmosphere to the MLT. There is indirect evidence (based on the phase relationship of the 27-day signatures in $H_2O$ and temperature) that the solar 27-day signatures in NLCs are driven by a 27-day modulation of the upwelling at the polar summer mesopause (Thomas et al., 2015).

An interesting discussion in the past decade is related to the 11-year solar signature in NLCs/PMCs. A solar influence is expected because mesopause temperature is positively correlated to solar activity, and the $H_2O$ abundance is anticorrelated to solar activity due to the photolysis of $H_2O$. The SBUV PMC record (e.g. DeLand and Thomas (2015)) – now covering about four 11-year solar cycles – does indeed show a pronounced 11-year signature until about the year 2002. During the last two decades, the solar effect is essentially absent. Hervig et al. (2019b) investigated potential reasons for the absence of the signature and concluded that a strongly reduced solar cycle signature in $H_2O$ at the polar summer mesopause since about 2000 is the main reason for the reduced solar response in NLCs. The underlying processes are not well understood and remain to be

investigated. The long-term evolution of stratospheric ozone and its recovery may be a potential contributor (Lübken et al., 2009). It has also been suggested that part of the apparent 11-year solar signature in PMCs in the SBUV record before 2000 may be caused indirectly by the eruptions of the volcanoes El Chichón in 1982 and Mount Pinatubo in 1991 (Hervig et al., 2016). The underlying physical mechanisms are not well understood, and may also involve intra- and inter-hemispheric coupling via gravity waves (Wallis et al., 2023).

## 3.4 Effects of energetic particle precipitation

It is important to note that solar photons are not the only energy source in the MLT. Energetic particles (mainly protons and electrons) that are emitted from the Sun during coronal mass ejections or are lost from the magnetosphere during geomagnetic storms can penetrate into the MLT and even the stratosphere at high (geomagnetic) latitudes. Precipitation of these particles leads to ionization of the major constituents in the MLT and the subsequent ion-molecule chemistry converts $H_2O$ to $HO_x$ and $N_2$ to $NO_x$ (Sinnhuber et al., 2012). Global observations of NO have been widely used to study particle impact ionization in the MLT region, e.g., showing the clear effect of particle forcing well into the mesosphere (e.g. Hendrickx et al. (2015); Hendrickx et al. (2017); Kiviranta et al. (2018); Kirkwood et al. (2015); Sinnhuber et al. (2016); Sinnhuber et al. (2022); Smith-Johnsen et al. (2018)). A similar response to particle forcing has been found for OH, based on observations from MLS/AURA in the high-latitude mesosphere (Andersson et al., 2012; Fytterer et al., 2015). A response of mesospheric NO to geomagnetic storms and auroral substorms has also been observed by a ground-based microwave radiometer (Newnham et al., 2018; Newnham et al., 2011).

Both $HO_x$ and $NO_x$ destroy $O_3$ in catalytic cycles. Development of 3-dimensional global chemical-dynamical models (Kovacs et al., 2016; Verronen et al., 2016) has enabled satisfactory simulations of the observed increases in $HO_x$ and $NO_x$ within the *E*- and *D*-region ionospheres and the subsequent decrease in mesospheric $O_3$, which may be depleted almost entirely in specific altitude ranges (Andersson et al., 2014; Nieder et al., 2014; Smith-Johnsen et al., 2018).

While the particle fluxes and atmospheric response during intense but sporadic Solar Proton Events (SPEs) has been well characterized (Funke et al., 2011; Jackman et al., 2009; Jackman et al., 2014), the continuous flux of medium energy electrons from the radiation belts and their effects on the atmosphere has been more difficult to quantify. This is mainly related to the lack of direct observations of the radiation belt electron precipitating fluxes, with current estimates mostly based on the relatively sparse MEPED datasets (Nesse Tyssøy et al., 2022; Peck et al., 2015; Pettit et al., 2021; Smith-Johnsen et al., 2018). Several studies have also investigated the upper mesospheric and thermospheric temperature response to geomagnetic storms and SPEs e.g. Nesse Tyssøy et al. (2010), Li et al. (2018), and Wang et al. (2021). The short-lived $HO_x$-driven $O_3$ depletion in the mesosphere should lead to reduced solar diabatic heating, although this cooling has never been directly confirmed by measurements. The temperature changes driven by SPEs affect the propagation of gravity waves and may lead to significant perturbations of the temperature at the polar summer mesopause (e.g. Krivolutsky et al. (2006); Becker and von Savigny (2010)). A strong warming at the southern hemisphere polar summer mesopause and a corresponding reduction in the occurrence rate of NLCs was observed during the January 2005 SPE (von Savigny et al., 2007). The interpretation of this

warming and the contribution of the SPE are difficult to diagnose because of enhanced quasi-2-day-wave activity that occurred at the same time (the quasi-2-day-wave is a planetary wave that is particularly significant in the mesosphere during summer) (Siskind and McCormack, 2014).

### 3.5 QBO and MJO signatures

Several studies have addressed the possible effects of the Quasi-Biennial-Oscillation (QBO) and the Madden-Julian-Oscillation (MJO) on the MLT region. Espy et al. (2011) provided experimental evidence for a QBO signature in OH(3-1) rotational temperature measurements at 60°N. The authors suggested that this signature is caused by a QBO-modulation of the interhemispheric coupling effect (Section 3.2). A QBO effect on the stability of the polar vortices – and the likelihood of a major stratospheric warming in the northern hemisphere – is well established (e.g., Labitzke (2004); Camp and Tung (2007)). Interhemispheric coupling provides a mechanism to transfer this signature from the polar winter stratosphere to the polar summer mesopause region. Effects of the MJO on the MLT have been addressed by only a handful of studies. The MJO is a dominant driver of intra-seasonal variability in the tropical region and is associated with the eastward (i.e. against the westward trade winds) propagation of a convective system (Madden and Julian, 1972). Several studies have investigated MJO effects on the stability of the Arctic polar vortex (e.g. Garfinkel et al. (2012); Yang et al. (2019)). A noted previously, the recent study by Hoffmann et al. (2022) suggests that the MLT region is also affected by the MJO via the interhemispheric coupling mechanism.

### 3.6. Long-term variations in the MLT region

Evidence for long-term variations in the MLT region is seen in different parameters. Since this topic has been the subject of recent reviews (e.g. Plane et al. (2015); Laštovička (2023)), it is briefly discussed here. Measurements of radio reflection heights (also known as standard phase heights) over Europe exhibit a long-term decrease at a rate of about 110 m decade$^{-1}$ between 1959 and 2009 (e.g. Peters et al. (2017)). This decrease is thought to be a consequence of the overall contraction of the middle atmosphere which is in part caused by increased radiative cooling by $CO_2$. Indeed, the rate of change is consistent with observed temperature changes in the middle atmosphere, as shown by Peters et al. (2017). Many studies have investigated long-term trends in middle atmosphere and MLT temperature. A long-term cooling trend in the middle mesosphere appears to be well established at a rate of about 3 K decade$^{-1}$ around 70 km; the trend in the mesopause region is much smaller, though possibly also negative (Beig, 2011).

### 4 Airglow Emissions and Mechanisms

During the day in the MLT, solar UV radiation excites optical transitions and photodissociates molecules. Various reactions, among which termolecular association of atomic O to form electronically-excited molecular $O_2$ has a prominent role, generate a plethora of atomic and molecular excited states. Many of these species are metastable, *i.e.*, their radiative lifetimes are

relatively long compared to the collision frequency with air molecules. Therefore, these excited species may emit radiation or
be deactivated by energy transfer in collisions with other atmospheric constituents. The resulting light emissions are collectively known as airglow, and have a global extent. The terms dayglow and nightglow are also used depending on whether the emission occurs during the day or night, respectively. Note that nightglow emissions are essentially only produced by chemiluminescent reactions, unlike dayglow where solar-stimulated emission is also important. Airglow is a common feature of planetary atmospheres (Krasnopolsky, 2011), and the spectra emanating from each planet's upper atmosphere vary widely
depending on its composition, temperature profile, and the incident solar flux. Accurate knowledge at the atomic and molecular level of the mechanisms that generate airglow is required for the interpretation of the observed emissions. When these details are well understood, remote-sensing observations can be used to retrieve a wealth of information on the composition, transport, winds, waves, and the energy balance of the upper atmosphere.

This section selectively highlights three important developments during the past decade relevant to our understanding of
nightglow emissions and their mechanisms: (1) recognition of the importance of multi-quantum vibrational-to-electronic energy transfer in collisions of highly vibrationally excited hydroxyl, OH(high v), with O atoms; (2) advances in understanding the nightglow from atomic Na and K, particularly the variability of the Na $D_2/D_1$ line ratio; and (3) the identification of continuum nightglow emissions from the metal oxides FeO and NiO.

### 4.1 Multi-quantum Relaxation in OH(v) + O Collisions

Light emitted from OH(v $\geq$ 5), known as the OH Meinel band emission, gives rise to some of the most prominent features of mesospheric nightglow in visible and near-infrared wavelengths. This emission is usually observed within a layer of approximately 10 km in width, peaking around 88 km. The main source of vibrationally excited OH is the reaction H + $O_3$ producing OH in its ground electronic state in vibrational levels v = 5–9 (Adler-Golden, 1997). This exothermic reaction is one of the major sources of heating in the MLT (Mlynczak et al., 1998), since the released chemical energy is redistributed by
collisions with other atmospheric gases into vibrational, rotational, and translational energy.

Remote sensing observations of the OH rotational temperature have been used for decades to measure the temperature around 88 km (Offermann et al., 2010). However, this is not straightforward: the OH(v) rotational temperature actually increases by approximately 15 K as v increases from 2 to 8, and v = 8 exhibits a significantly higher rotational temperature than OH(v = 9) (Cosby and Slanger, 2007). This discovery has stimulated recent work on OH rotational temperatures (e.g. Franzen et al.
(2020); Kalogerakis (2019b); Noll et al. (2017)) and Einstein coefficients (e.g. Hart (2021); Liu et al. (2015); Noll et al. (2020)).

Because of their atmospheric relevance, collisional processes involving OH(v) and the major components of the atmosphere at this altitude region, $O_2$ and $N_2$, have been studied for many years. However, atomic O forms a significant component of the atmosphere at the high-altitude part of the OH(v) layer, and detailed quantitative knowledge regarding collisional energy
transfer with O has been relatively limited. Important relevant developments in the past decade have been the realization that collisional loss of OH(v = 9) by O atoms is extremely fast, with a total removal rate constant determined at $(4 \pm 1) \times 10^{-10}$ cm$^3$

s$^{-1}$ at room temperature (Kalogerakis et al., 2011), and involves multi-quantum, vibrational-to-electronic (V–E) energy transfer (Kalogerakis et al., 2016; Sharma et al., 2015):

$OH(v = 9) + O(^3P) \rightarrow OH(v = 3) + O(^1D)$             (1)

Eq. 1 is almost thermoneutral and represents the most efficient loss process for OH(v = 9) by O atoms. Its rate constant is more than one order of magnitude larger than the reported rate constant value of $(3.3 \pm 0.5) \times 10^{-11} \, cm^3 \, s^{-1}$ for the reaction of OH(v = 0) with O atoms that yields hydrogen atoms and molecular oxygen. Moreover, Eq. 1 provides a source of nighttime O($^1D$)

atoms that are rapidly deactivated by the two main atmospheric constituents $N_2$ and $O_2$. As a result, the electronic energy of the O($^1D$) atom can be channelled to other excited states and becomes itself a driving force generating other nightglow emissions.

Deactivation of O($^1D$) by $N_2$ eventually populates $N_2(v = 1)$, which transfers vibrational energy to the $CO_2(v_3)$ antisymmetric stretch, resulting in an enhancement of the $CO_2(v_3)$ emission at 4.3-μm (Panka et al., 2017; Sharma et al., 2015). This pathway

provides a solution to the long-standing problem of unacceptably large discrepancies between observations and model calculations of the nighttime 4.3-μm emission (Panka et al., 2017). The Panka et al. paper was published in *Atmospheric Chemistry and Physics* in 2017, 43 years after the 1974 sounding rocket measurements which revealed something major was amiss in our understanding of the $CO_2$ 4.3 μm emission (Kumer et al., 1978).

The OH(v ≥ 5) + O multi-quantum vibrational relaxation process of Eq. 1 also efficiently transfers energy to $O_2$ because O($^1D$)

+ $O_2$ collisions generate $O_2(b^1\Sigma_g^+)$ in vibrational levels v = 0, 1 (Pejaković et al., 2014). This OH(v ≥ 5) $\rightarrow O_2(b^1\Sigma_g^+)$ energy flow pathway was recently shown to represent a significant, previously unrecognized source of the $O_2(b^1\Sigma_g^+ - X^3\Sigma_g^-)$ Atmospheric band emission, another prominent feature in Earth's nightglow spectrum (Kalogerakis, 2019a). In summary, these important recent advances demonstrate the central role of O atoms – more complex than previously recognized – in generating, regulating, and interconnecting multiple airglow emissions.

**4.2 Na and K nightglow emission**

Although Na D-line emission from the upper atmosphere was first reported in the 1920s, a fairly recent and surprising discovery is that the ratio ($R_D$) of the two lines at 589.0 and 589.6 nm varies between about 1.5 and 2.0 (Harrell et al., 2010; Plane et al., 2012; Slanger et al., 2005). This variability cannot be explained by the original Chapman mechanism for Na D emission, which proposed that NaO, produced from the reaction Na + $O_3 \rightarrow$ NaO + $O_2$, reacts with O to release the Na atom

in the excited $^2P$ state with a fixed propensity of the spin-orbit states. However, a laboratory experiment simulating the nightglow chemistry showed that $R_D$ varied when the ratio of [O]/[$O_2$] was varied over the range typical of the mesosphere (Slanger et al., 2005). This led to a modified Chapman mechanism being proposed: Na + $O_3$ makes excited NaO exclusively

in its first excited electronic state ($A^2\Sigma$), which can then react with O to make $Na(^2P)$ with the $j = 3/2$ and 1/2 states in the ratio 2:1; or, the $NaO(A^2\Sigma)$ can be quenched by $O_2$ to ground-state $NaO(X^2\Pi)$, with a $j$ ratio of 1:5 (Plane et al., 2012).

Recently, extensive K nightglow observations were reported using the Ultraviolet and Visual Echelle Spectrograph at Cerro Paranal in Chile (Noll et al., 2019). This study showed that the K nightglow, which is generated by the reaction between K and $O_3$, has a quantum yield of ~30%, which is much larger than the quantum yield of ~6% for Na(D) emission (Unterguggenberger et al., 2017).

### 4.3 Identification of Atmospheric Metal Oxide Emissions

Nightglow spectra measured by both OSIRIS (Evans et al., 2010), the Keck II telescope in Hawaii (Saran et al., 2011), and the Very Large Telescope in Chile (Unterguggenberger et al., 2017) show that underlying the strong nightglow features between 540 and 680 nm from O, OH and Na there is a quasi-continuum with maximum intensity at 595 nm. This is produced by electronically excited FeO, formed by the reaction $Fe + O_3 \rightarrow FeO^* + O_2$, with a quantum yield of about 13%. The same emission band was observed in the spectrum of a persistent Leonid meteor train (Jenniskens et al., 2000). An analogous –
though much fainter – emission from electronically excited NiO has been detected at wavelengths longer than 440 nm in spectra taken by OSIRIS and the GLO-1 instrument flown on the Space Shuttle (Evans et al., 2011). The quantum yield for the $Ni + O_3 \rightarrow NiO^* + O_2$ reaction is similar to that of Fe (Daly et al., 2020).

### 5 Meteoric Metals

Meteoric ablation produces layers of metal atoms and ions that occur globally between about 80 and 120 km. Besides the
curiosity of these highly reactive species existing in an ostensibly oxidizing atmosphere, several of the metals (Na, Fe, K, Ca and Ni) can be observed with high temporal and spatial resolution from the ground by the resonance lidar technique (Section 2.3). The metals therefore provide an important tool for studying the chemistry and physics of the MLT. In the lower thermosphere the metals occur as atomic ions, and these are the principal component of sporadic $E$ layers which affect radio propagation (Plane, 2003). In this Section we discuss advances in understanding meteoric ablation, the chemistry of the
meteoric metals, formation of meteoric smoke particles, sporadic $E$ layers, and the long-running question of the magnitude of the cosmic dust influx into the atmosphere.

### 5.1 Meteoric ablation

Cosmic dust particles enter the Earth's atmosphere with velocities of between 11 and 72 km s$^{-1}$ (Plane et al., 2015). At these hypersonic velocities, inelastic collisions of a dust particle with air molecules cause a small mass loss through sputtering;
however, if the particle heats to a temperature above ~1800 K it will melt, resulting in the rapid ablation (i.e. evaporation) of the relatively volatile constituents (Na and K), before the major elements (Fe, Mg and Si) ablate around 2000 K. If the residual particle heats to ~2400 K, the refractory elements (Ca, Al and Ti) ablate. In the past decade, two types of apparatus have been

developed to study ablation of micron-sized particles. The first of these employs programmed flash heating of micron-sized meteoritic particles on a filament, with the time-resolved evaporation of pairs of metal atoms (e.g. Na and Fe) monitored by fast time-resolved laser induced fluorescence (Bones et al., 2016). A second version of this meteoric ablation simulator was developed recently to study the pyrolysis of the organic material that binds the mineral grains in meteorites; this used fast time resolution mass spectrometry to observe oxidation products such as $CO_2$ and $SO_2$ following the deposition of meteoritic particles on a hot surface (Bones et al., 2022). The second apparatus is an electrostatic dust accelerator, which is used to generate pure metal (e.g. Fe) particles or poly-pyrrole-coated silicate particles with velocities of 10 - 70 km s$^{-1}$. The particles are then introduced into a chamber pressurized with a target gas, and ablation monitored using an array of electrodes to probe the resulting impact ionization of the ablated metallic atoms (DeLuca et al., 2022; Thomas et al., 2017).

The results of these laboratory studies have been used to test and refine a chemical ablation model (CABMOD), which describes the ablation of individual elements from a cosmic dust particle with a specified mass, velocity and entry angle into the atmosphere (Gómez Martín et al., 2017a). This model incorporates both a silicate and an Fe-Ni-S phase (Carrillo-Sánchez et al., 2020), and recently an organic phase has been included (Bones et al., 2022). CABMOD has proved to be useful in two ways. First, the model has been used to understand and interpret high performance meteor radar observations (Section 2.3), since it can be used to predict the rate of generation of electrons, via collisional ionization of the ablating metal atoms; it is these electrons that cause incoherent radar scatter from the resulting plasma around the ablating particle (Janches et al., 2017). Second, CABMOD has been coupled to an astronomical dust model ZoDY (Nesvorný et al., 2011), which predicts the size and velocity distributions of dust particles from cometary and asteroidal sources that are incident on the Earth's atmosphere. The CABMOD-ZoDY model can then be used to predict the injection rates as a function of altitude of individual meteoric metals into the Earth's atmosphere, as well as other planets (Carrillo-Sánchez et al., 2020).

## 5.2 Metal chemistry

The substantial increase in the database of metal atom and ion observations (Section 2.3) has been complemented by advances in laboratory studies of neutral and ion-molecule reactions of metallic species (Plane et al., 2015). Notable new techniques include: the development of pulsed laser photo-ionization/time-of-flight mass spectrometry to study neutral metallic compounds which do not have suitable optical transitions e.g. for the reactions of species like NaOH (Gómez Martín et al., 2017b), CaOH and $O_2CaOH$ (Gómez Martín and Plane, 2017) with H and O; a discharge flow reactor for studying dissociative recombination reactions of metallic molecular ions such as $FeO^+$ and electrons (Bones et al., 2015); and a selected ion flow tube (SIFT) apparatus for measuring the product branching ratios of reactions such as $FeO_x^+$ ($x = 0 - 4$) + $O_3$ (Melko et al., 2017). These laboratory rate coefficients, together with high level *ab initio* quantum theory calculations and statistical rate theories to determine reaction mechanisms and estimate rate coefficients for reactions that have not been studied experimentally, have enabled comprehensive reaction schemes for Na, K, Fe, Mg and Ca (Plane et al., 2015), and more recently Ni (Daly et al., 2020) and Al (Plane et al., 2021), to be developed.

These reaction schemes can then be used in atmospheric models. Although 1-dimensional models of the meteoric metal layers have been used for several decades (see Plane (2003) for a review), it is only in the last decade that global 3-dimensional modelling has been performed. In particular, the Whole Atmosphere Community Climate (WACCM) model, developed at the National Center for Atmospheric Research (Boulder, Colorado), has been used to model the Na (Marsh et al., 2013), Fe (Feng et al., 2013), Ca (Plane et al., 2014a), Mg (Langowski et al., 2015) and Ni (Daly et al., 2020) layers up to ~140 km. The

injection rate profile for each metal is supplied by the CABMOD-ZoDY model (Section 5.1). Generally, very satisfactory agreement with observations (by lidar, satellite and rocket) is achieved, if the metal injection rates are divided by a factor around 5. This accounts for the fact that WACCM (and other global models) underestimate the vertical transport of minor species in the MLT: this appears to be because short wavelength gravity waves are not resolved on the model grid, and these sub-grid waves contribute to vertical chemical and dynamical transport of constituents while dissipating (Gardner et al., 2017).

The WACCM model has successfully simulated the effect of a stratospheric warming on the Na and Fe layers (Feng et al., 2017), and been used to explore the effect of the solar cycle and long-term change on several metal layers (Dawkins et al., 2016).

The increase in thermospheric lidar observations (Section 2.3) has stimulated model development in very high-top models, also driven by an increased interest in the global occurrence of sporadic $E$ layers, which are narrow layers of high

concentrations of mainly $Fe^+$ and $Mg^+$ ions and electrons (Yu et al., 2021). The extended version of WACCM (WACCM-X, with a top around 500 km) has recently been modified to treat the electrodynamical transport of metallic ions in the Earth's magnetic field (Wu et al., 2021). The global ionospheric transport of metallic ions has also been explored in the SAMI3 model (Huba et al., 2019). Chu and Yu (2017) have developed a regional model (TIFe) to explain the formation of thermospheric Fe layers over Antarctica.

**5.3 Meteoric smoke particles**

Fe, Mg and Si should be the major elements injected into the MLT from meteoric ablation, with roughly similar rates (Carrillo-Sánchez et al., 2020). Laboratory studies have shown that when these elements are oxidized in the presence of $O_3$, $O_2$ and $H_2O$, they then polymerize into nm-size aerosol particles of olivine ($Fe_{2x}Mg_{2(1-x)}SiO_4$, x = 0 – 1) or pyroxene ($Fe_xMg_{(1-x)}SiO_3$, x = 0 – 1) composition (Saunders and Plane, 2011). This is likely to be what occurs to the ablated metals below 85 km where the

atmosphere becomes strongly oxidizing.  The resulting meteoric smoke particles (MSPs) have been detected as charged particles by rocket-borne instruments (Section 2.2). Relatively heavy charged MSPs also cause a distinctive line shape in incoherent scatter radar spectra, as a result of different diffusion modes in the $D$ region plasma (Strelnikova et al., 2007). Although attempts to collect MSPs in the mesosphere using the rocket-borne MAGIC system have been unsuccessful (Hedin et al., 2014), the optical extinction of MSPs has been measured using the SOFIE spectrometer on the AIM satellite, and multi-

wavelength extinction measurements show that the composition is consistent with Fe-Mg-silicates (Hervig et al., 2017; Hervig et al., 2021). Aircraft-based observations of meteoric material in lower-stratospheric aerosol particles between 15 and 68° N have provided important new information about the transport of MSPs down the winter polar vortex and their subsequent

distribution to low latitudes by isentropic mixing, as well as indicating a substantial input of meteoric fragments (Schneider et al., 2021).

Recent laboratory work has played an important role in understanding the microphysics and chemistry of MSPs: determination of the optical properties of MSP analogues, which is important for atmospheric observations (Aylett et al., 2019); confirmation that meteoric smoke particles can provide nuclei for heterogeneous ice formation in NLCs (Duft et al., 2019); the effect of solar radiation on ice particle formation (Nachbar et al., 2019); demonstration that MSPs provide an active surface for heterogeneous chemistry e.g. altering the nighttime $HO_2$/OH ratio (James et al., 2017); and that MSP analogues and meteoritic

fragments are effective heterogeneous nuclei for crystal formation of nitric acid trihydrate (NAT) in the absence of water ice in polar stratospheric clouds (James et al., 2023).

## 5.4 Magnitude of the cosmic dust flux

In a review a decade ago, Plane (2012) showed that estimates of the input rate of cosmic dust – measured by space-based, atmospheric and ground-sampling techniques – ranged from ~3 to 300 tonnes per day globally. This 2 order-of-magnitude

uncertainty has been narrowed considerably since then. Making use of the measured vertical fluxes of Na and Fe in the upper mesosphere (Section 2.3), and the accumulation flux of cosmic spherules at the South Pole, Carrillo-Sánchez et al. (2020) used the CABMOD-ZoDY model to obtain an estimate of $28 \pm 16$ t d$^{-1}$. This is in excellent agreement with an estimate of $25 \pm 7$ t d$^{-1}$ from SOFIE-AIM optical extinction measurements (Hervig et al., 2021), and $22 \pm 13$ t d$^{-1}$ from the Wind satellite *in situ* dust detection network (Hervig et al., 2022). The CABMOD-ZoDY model flux was further validated against an important

collection of cosmic spherules and micrometeorites from Concordia, Antarctica (Rojas et al., 2021). The remaining discrepancy in the dust input rate is the very much higher estimates (> 200 t d$^{-1}$) from the accumulation of MSPs in polar ice cores, which Brooke et al. (2017) showed in a global modelling study cannot be explained by focusing effects during atmospheric transport from the upper mesosphere to the surface.

## 6 Ice Clouds in the Mesosphere

NLCs remain a very vital field of middle atmosphere research. This concerns not only the properties and lifecycle of the clouds, but there is also an increasing focus on using these clouds as a convenient observable for studying MLT processes from a more general perspective. A recent review on NLC/PMCs has been provided by von Savigny et al. (2020). The clouds occur in the upper mesosphere at high latitudes during summer, where 24 hours of sunlight provide continuous solar heating and photolysis of water vapour in the ultraviolet. The fact that clouds can form under these conditions is another strong manifestation of the

wave-driven circulation that governs the mesosphere on a global scale. The strong upwelling connected to this circulation in the polar summer provides both the extreme adiabatic cooling and the efficient transport of water vapour from below that make the clouds possible.

## 6.1 Cloud microphysics

Over recent decades there have been major developments in NLC observational capabilities, ranging from long-term and increasingly detailed lidar studies (e.g., Fiedler et al. (2017); Ridder et al. (2017); Kaifler et al. (2023)) to satellite missions addressing properties of the clouds in direct connection to their mesospheric environment (e.g., Rong et al. (2012); Hervig et al. (2015); Christensen et al. (2016)). At the same time, a quantitative understanding has developed of the clouds' critical sensitivity to their atmospheric environment. This forms the basis for now using NLCs as an observational tool to investigate a wide range of processes ranging from small scale dynamics (Fritts et al., 2019; Kaifler et al., 2023) via characterisation of gravity waves (Thurairajah et al., 2013; Zhao et al., 2015) to global coupling processes (Gumbel and Karlsson, 2011; Karlsson et al., 2007).

From a chemistry perspective, the nucleation and growth of mesospheric ice particles is of particular interest. Today, it is well established that NLCs consist of water ice, albeit containing considerable amounts of meteoric material (Hervig et al., 2001; Hervig et al., 2012). NLCs are observed when temperatures fall below about 145 K, and heterogeneous nucleation is deemed necessary for their formation under typical mesospheric conditions. As for condensation nuclei, MSPs and heavy ion clusters have long been considered as the two main candidates. In fact, these two categories of particles are not completely distinct as MSPs are subject to charging in the mesosphere, which can increase their efficiency as condensation nuclei (Gumbel and Megner, 2009). Charging mechanisms for MSPs comprise capture of electrons and positive ions, photo-ionization and/or electron detachment by solar photons, as well as secondary electron emission caused by energetic particle impact (Baumann et al., 2015; Baumann et al., 2016; Baumann et al., 2013). During the night, charge capture generally dominates, with more efficient capture of mobile free electrons compared with heavier ions. This causes a substantial fraction of MSPs to be negatively charged (Plane et al., 2014b). During daytime, on the other hand, solar effects usually dominate, leading to a substantial fraction of MSPs being positively charged. Indeed, rocket-borne measurements have revealed mesospheric profiles of meteoric smoke both with dominating negative and dominating positive charges (Baumann et al., 2013). The positive charging condition can be expected to prevail in the constantly sunlit polar summer mesosphere, i.e. the region where NLCs are nucleated. A major uncertainty remains the behaviour of the smallest MSPs with regard to the above charging processes. As these particles fall in a size regime between molecular properties and bulk particle properties, quantitative knowledge about charging efficiencies is sparse (Megner and Gumbel, 2009). In any case, nucleation of water ice on MSP-analogue particles only around 1 nm in radius has been demonstrated in the laboratory (Duft et al., 2019), so that charged MSPs do not seem to be a necessary requirement for NLC formation.

In addition to the heterogenous nucleation of ice particles described above, the occurrence of homogeneous nucleation has been considered: Zasetsky et al. (2009) and Murray and Jensen (2010) suggested pathways involving initial formation of amorphous solid water, followed by conversion into cubic or hexagonal ice. This path may be competitive at extremely cold conditions (~110 K), in transient local temperature minima caused by passing gravity waves.

## 6.2 Trends in mesospheric clouds

There has been a long-standing debate about whether NLC/PMCs can be regarded as an indicator of anthropogenic climate change in the middle atmosphere (Thomas, 1996; von Zahn, 2003). In general, reasons considered for enhanced NLC formation comprise both cooling of the middle atmosphere due to increasing concentrations of $CO_2$, and increasing humidity in the middle atmosphere produced by oxidation of the increasing concentrations of methane (this accounts for about half of the amount of water vapour in the middle atmosphere). Trends in the cloud brightness and cloud ice water content have been deduced from satellite observations (e.g., DeLand and Thomas (2015)), but time series are affected by both solar variability and changes in stratospheric ozone, so the statistical significance of long-term trends remains limited.

As discussed in Section 3.6, the middle atmosphere has indeed been cooling significantly over recent decades as a consequence of enhanced $CO_2$ as the major radiative cooling agent above the tropopause (Goessling and Bathiany, 2016). However, as it turns out, the polar summer mesopause region is an exception to this general cooling trend. Here, the strong adiabatic cooling due to the mesospheric circulation, rather than thermal emission from $CO_2$, is the major cooling process causing the very low temperatures in summer. At these very low temperatures, thermal emissions are small and variations of $CO_2$ have little effect on the overall cooling rate. In fact, under these conditions an important radiative effect of $CO_2$ at the summer mesopause is the absorption of thermal radiation from the stratosphere. As overall $CO_2$ concentrations and thus the optical thickness in the centre of the 15-μm $CO_2$ absorption band increase, this radiation originates from higher and warmer stratospheric altitudes, leading to an increased contribution to heating near the summer mesopause. The identification of trends in NLCs is further complicated by a long-term shift in their altitude: the $CO_2$-induced cooling throughout the middle atmosphere below the clouds has caused a contraction of the atmosphere. As a consequence, while the atmospheric pressure level at which NLCs occur has not changed much, this pressure level has been moving to lower geometric altitudes (Bailey et al., 2021; Lübken et al., 2013). Based on detailed model studies, Lübken et al. (2021) conclude that at a given pressure level long-term temperature trends are very small in the summertime upper mesosphere.

In summary, a role of NLCs as an indicator of anthropogenic climate change in the middle atmosphere is certainly not straight-forward, considering the combined effect of changes in temperature, water vapour and pressure level altitudes. Nonetheless, as described in Section 6.1, the fact remains that NLC are a very convenient observable for a wealth of other scientific issues in the mesosphere, covering a wide range of temporal and spatial scales. From a chemical perspective, variations in NLCs are expected to influence the mesosphere in various ways. The growth, sedimentation and subsequent sublimation of NLC particles leads to a dehydration near the mesopause and a transport of water to lower altitudes (Hervig et al., 2015). This results in substantial effects on the local $HO_x$ and $O_x$ chemistry (Murray and Plane, 2005; Siskind et al., 2018). Lübken (2022) suggests additional effects in other parts of the middle atmosphere as absorption of solar ultraviolet radiation by NLCs may lead to changes in photochemical processes at lower altitudes.

## 7 Future Directions

In this Section, we suggest future directions for research into the chemistry of the MLT. As might be expected, most of these suggestions involve currently unsolved problems, or the use of new observational platforms and techniques; a few directions are more speculative. We also flag up potential gaps in observational capabilities over the next decade. To some extent the discussion reflects the backgrounds and research interests of the authors, and is not meant to be an exhaustive list.

### 7.1 Observations

Many current satellite missions with a focus on MLT chemistry are approaching the end of their lifetimes. There are serious concerns regarding a lack of planned space missions to continue the important task of monitoring this part of the atmosphere and its complex roles in the climate system and solar-terrestrial coupling (Mlynczak et al., 2021). Nevertheless, there are some new satellite missions recently launched or in the pipeline. The Swedish MATS satellite (Mesosphere Airglow/Aerosol Tomography and Spectroscopy) was launched in November 2022. Its main science objective is the study of gravity waves and their interactions in the MLT (Gumbel et al., 2020). Based on limb imaging and tomography of $O_2$ airglow and NLCs, MATS will provide global three-dimensional fields of temperature and related quantities as a key to wave analysis. The Atmospheric Waves Experiment (AWE) is NASA's first space mission dedicated to mesospheric gravity waves (https://blogs.nasa.gov/awe). AWE is due to be mounted on the International Space Station in late 2023. Measurements are based on a nadir-viewing temperature imager from Utah State University, utilizing OH Meinel nightglow emissions (Pautet et al., 2014).

Fortunately, data about the MLT can become available from space missions dedicated to other parts of the atmosphere. The Belgian mission ALTIUS (Atmospheric Limb Tracker for Investigation of the Upcoming Stratosphere) is currently scheduled for launch in 2025 and will also provide some capability to observe and investigate NLCs (https://www.esa.int/Applications/Observing_the_Earth/Altius). Under evaluation as a future ESA Earth Explorer mission is The Changing-Atmosphere IR Tomography Explorer (CAIRT) aiming at measurements of temperature and numerous trace gases from the tropopause to the lower thermosphere (https://www.cairt.eu). From a more general perspective, it will be crucial to consolidate and extend the long-term NLC/PMC data record that is based on nadir backscatter measurements in the UV and that has its origin in the series of SBUV satellite instruments. The JPSS-2 (Joint Polar Satellite System 2) satellite was launched in November 2022, hosting the OMPS instrument that allows the retrieval of PMC albedo and ice water content (DeLand and Thomas, 2019). The continuation of these measurements is guaranteed well beyond 2030, based on the planned launches of the JPSS-3 (2027) and JPSS-4 (2032) missions (https://www.nesdis.noaa.gov/about/our-offices/joint-polar-satellite-system-jpss-program-office).

As for sounding rockets, there has been a decline in the number of projects compared with the 1980s and 1990s. Three main reasons can be identified: many questions needing sounding rocket experiments appear to have been answered; there has been remarkable progress in active ground-based lidar and radar instruments that provide detailed and continuous information on

relevant scales; and there has been growing competition for funding, not least from cubesats and other emerging small satellite technologies. Nonetheless, there is certainly a continued need for sounding rockets to address the intimate local interplay of

physical and chemical processes in the MLT.

Indeed, there are intriguing new ideas regarding sounding rocket experiments. An important aim is to go beyond the limitation of one-dimensional profile measurements traditionally delivered by sounding rockets. New techniques use the ejection of secondary instrument modules from a primary rocket payload, thereby enabling three-dimensional multi-profile measurements. As for composition measurements, new generations of rocket-borne ion mass spectrometers can be expected

to bridge the gaps between molecular ions, clusters and charged particles (Stude et al., 2021). New ideas have also emerged for measuring atomic O based on its fine structure emission at 63 μm in the terahertz spectral region (Richter et al., 2021; Yee et al., 2021). An issue that remains to be addressed is the sampling of gaseous species from sounding rockets.

Ground-based measurements are also taking important steps beyond traditional "one-dimensional" views of the atmosphere. Regarding radar studies of the MLT, phased-array systems allow for increasing flexibility and the probing of processes and

structures in three dimensions (e.g., Latteck et al. (2010)). In Northern Europe, EISCAT_3D will soon provide a powerful radar system closely connecting research on solar-terrestrial interactions, upper atmosphere and the MLT (https://www.eiscat.se). An important way of adding horizontal components to MLT measurements is the application of passive and active instrumentation on aircraft or balloons (e.g. Kaifler et al. (2023); Pautet et al. (2019)).

Connecting instrumentation in comprehensive networks can add a valuable global component to ground-based measurements.

Currently the most ambitious network is the Meridian Space Weather Monitoring Project of the Chinese National Space Science Centre (Chi et al., 2020). This network comprises 15 monitoring stations along 30ºN and 120ºE, equipped with magnetometers, ionosondes and digisondes, radars, lidars, Fabry-Perot interferometers and sounding rockets (https://data.meridianproject.ac.cn). Other examples include the networking of meteor radars to derive large-scale wind fields in the MLT (e.g. Stober et al. (2021)), and the Midlatitude Allsky-imaging Network for GeoSpace Observations (MANGO),

a collection of all-sky cameras and Fabry-Perot Interferometers spread over the continental US to image large-scale airglow and aurora features from which neutral winds and temperatures can be determined (https://www.mangonetwork.org). An ultimate goal of both ground-based and space-borne measurements remains near-simultaneous global coverage, with data assimilation (DA) extending to the MLT and beyond (e.g., Eckermann et al. (2018)). For this reason, it is important to continue supporting geophysical observation facilities, particularly bearing in mind the loss of major facilities such as Arecibo and

Sondrestrom in the past decade.

## 7.2 Atmospheric model developments

Over the last two decades we have seen the development of "whole atmosphere" 3-dimensional models such as WACCM, HAMMONIA, EMAC and KASIMA that span the MLT and include comprehensive interactive chemistry (Sinnhuber et al., 2022). There are a number of directions that future model development can be expected to take. First, there will be a push to

increase the horizontal and vertical resolution of these models to better resolve the full spectrum of waves in the atmosphere

and to capture across-scale coupling processes (e.g., the interaction of tides and gravity waves), and to reduce the need to parameterize small-scale processes. Global simulations at high resolution have been conducted for climate research (Chang et al., 2020; Hohenegger et al., 2023) and for studies of mesospheric gravity waves (Liu et al., 2014). However, increasing resolution globally comes at a large numerical cost: a factor of 4 to 8 in cost for each doubling of the resolution. To overcome this, models that increase resolution on only part of the domain (so-called "regionally-refined" models) are under development, which would, for example, have a region of 1/8° horizontal resolution within a global 1° grid. Again, such models have been developed but only applied in simulations of the troposphere (Herrington et al., 2022; Schwantes et al., 2022). Regionally-refined models allow for the two-way coupling of planetary-scale and small-scale variability but do require that new unstructured grids be used, which greatly complicates analysis of model output. It is expected that insights gained from such models will lead to improvements in the parameterization of sub-scale processes used in coarser-scale models.

Second, it will likely become common practice to run ensembles of model simulations to capture the spread of possible atmospheric states that comes from geophysical variance (Richter et al., 2022). The MLT has weather, just as in the troposphere, and its evolution in time is not solely determined by external forcing either from above or below. As with any chaotic system, small changes in initial conditions can lead to large changes in the final model states (Liu et al., 2009). Such ensembles can potentially be used to make forecasts of the MLT on timescales from hours to weeks, benefitting forecasting for space and tropospheric weather as well as sectors that rely on GPS and radio communications.

Third, DA systems will be used to improve the initialization of forecasts, to create better estimates of atmospheric state variables that are not easily measured, and to identify model biases. To date, these systems have almost exclusively assimilated dynamical quantities (McCormack et al., 2021), but extensions are now underway as DA systems evolve to include the assimilation of constituents and radiances/airglow (Eckermann et al., 2018). Unfortunately, the foreseeable lack of MLT satellite missions (Section 7.1) will challenge these efforts. DA systems can also be used to investigate complex processes: for example, while solar 11-year and 27-day signatures in MLT parameters are relatively easy to detect – if sufficiently long time series are available – the contribution of the different relevant processes are difficult to quantify and the level of understanding of many of the identified solar signatures in MLT parameters is quite low; this will be an important topic for future investigations.

Finally, since many of the whole atmosphere models began as climate models, extended simulations should be conducted to evaluate past decadal trends and possible future states of the MLT. Of particular interest will be the consequences of continued global warming and possible climate intervention actions, such as solar radiation management or other forms of geo-engineering.

## 7.3 Airglow Mechanisms

Atomic O plays a quintessential role in the photochemistry of the MLT region. Nevertheless, reactive and energy transfer processes involving O atoms are some of the least well understood and quantified, despite their critical role in airglow mechanisms.

Good knowledge of atmospheric composition is a prerequisite for any attempt to interpret and model airglow observations. The number densities of atomic O and H have been challenging to measure routinely. There has been some encouraging progress in the past decade, involving the use of solid electrolyte sensors for the determination of O atoms in sounding rocket experiments (Eberhart et al., 2019). However, there is still much to be desired. Improvements in remote-sensing capabilities for both O and H would be extremely helpful (Section 7.1): combined with simultaneous observations of multiple airglow emissions, they would provide stringent consistency checks on how well we understand the relationship between airglow emission intensities and atmospheric composition.

The recently established role of O atoms coupling the OH Meinel with the $CO_2$ 4.3 μm and the $O_2$ Atmospheric band emissions (Section 4.1) indirectly highlights the limitations in our understanding of the process of O + O termolecular association and the corresponding yields of $O_2$ electronically excited states contributing to airglow. Prior to that work, O atom termolecular association was thought to adequately account for the generation of electronically excited $O_2$ states in the nightglow. Concerted observational, theoretical, modelling, and laboratory efforts are required to address this deficiency in the coming years. At the same time, it seems we have barely scratched the surface regarding the quantitative details of the OH(v = 5 - 9) + O multi-quantum energy transfer pathway shown in Eq. 1. The vibrational quantum level dependence, as well as the temperature and collision energy dependence for this process are not well known, yet they are required parameters for accurate atmospheric modelling of multiple airglow emissions.

Despite numerous studies, the process of $CO_2$ vibrational excitation by O atoms remains poorly understood and represents a major unresolved challenge in upper atmospheric science. Unacceptably large discrepancies by factors of 3 - 8 exist between laboratory rate constant determinations for O-atom excitation or relaxation of $CO_2(010)$, and the corresponding values retrieved from atmospheric observations. Furthermore, the rate constants retrieved from observations exhibit an altitude dependence (Feofilov et al., 2012) that contradicts the temperature dependence reported by state-of-the-art laboratory experiments (Castle et al., 2012). Chemistry-climate models tend to use a median value for the excitation rate constant that probably does not reflect the actual value for this process. These discrepancies indicate a fundamental deficiency in our understanding of the underlying processes, or other flaws in the current treatment of non-local thermodynamic equilibrium in upper atmospheric models of the terrestrial planets.

## 7.4 Cosmic dust

As discussed in Section 5, significant progress has been made in several areas: determining dust sources and the magnitude of the dust flux entering the Earth's atmosphere; quantifying the injection profiles of individual elements from meteoric ablation; studying new species (Ni); extending lidar observations of metals into the thermosphere; and developing global models to describe the chemistry of 10 elements produced by meteoric ablation (Si , S and P in addition to the metals). Nevertheless, important uncertainties remain. The injection fluxes of the elements in a global chemistry-climate model such as WACCM need to be reduced by a factor of ~5 in order to simulate the *absolute* concentrations of the metals (Na, K, Fe, Mg, Ca and Ni) that are observed from the ground or from space. One plausible explanation for this is that vertical transport of the metallic

species down to below 80 km is underestimated because general circulation models do not have sufficient horizontal resolution to capture short-wavelength gravity waves that dissipate in the MLT (Section 5.2). This problem may be resolved by working with regionally-refined models (Section 7.2), and/or parameterising dynamical transport in lower-resolution models (Gardner, 2018). Another unsolved problem is that the MSP accumulation flux estimated from ice cores in both polar regions – using measurements of Ir, Pt and superparamagnetic Fe particles – is much higher than estimates of the ablation flux in the MLT (Section 5.4). As discussed in Brooke et al. (2017), there are no convincing explanations for this discrepancy in terms of a focusing effect produced by atmospheric circulation between the upper mesosphere and the troposphere, and wet/dry surface deposition. Future work should probably combine different techniques for analysis of ultra-trace species in ice cores, and include ice cores from glaciers located at low latitudes.

Although a recent study of the pyrolysis of refractory organics in meteoritic fragments suggests that fragmentation of cosmic dust particles smaller than 100 μm radius is unlikely (Bones et al., 2022), there are several compelling reasons for further study of fragmentation. These include: interpreting reports in studies using high performance meteor radars that fragmentation is commonplace (e.g. Malhotra and Mathews (2011)); understanding the effect of fragmentation in potentially reducing the total ablation rate from a population of cosmic dust particles; the importance of meteoritic fragments in PSC freezing in the polar winter stratosphere (James et al., 2023); and the possible role of fragments in skewing measurements of MSPs in ice cores.

In terms of the future development of chemical networks for models of the meteoric metals, an important goal of future laboratory work should be kinetic studies of reactions involving metallic compounds beyond the metal atoms and oxides that can be monitored by time-resolved laser induced fluorescence spectroscopy (Plane et al., 2015). Although there has been recent progress in developing a photo-ionization mass spectrometric technique to monitor species like NaOH which do not have suitable optical transitions (Gómez Martín et al., 2017b), more work needs to be done in this area.

Recently, a global Al model has been used to predict the density of AlO in the MLT (Plane et al., 2021), because this species has a strong laser fluorescence band around 480 nm and so should be observable by lidar. Lidar observations reported in this study determined an upper limit for AlO of only 60 cm$^{-3}$, not far above the model simulation of a nighttime concentration of around 10 cm$^{-3}$ (Plane et al., 2021). AlO should be a target for future lidar work, along with continuing to extend observations of metal atoms into the middle thermosphere (Section 2.3), where a number of surprising phenomena have already been observed (e.g. Chu et al. (2020)).

An important goal remains to pinpoint the composition and structure of MSPs. This will ultimately require rocket-borne sampling of the particles in the MLT, either directly or in the form of meteoric residuals in mesospheric ice particles. The development and characterisation of an appropriate sampling technique remains an open issue. Sampling of MSPs in the stratosphere above 30 km (i.e. higher than the Junge sulphate particle layer) would also provide important information about the chemical processing (aging) of MSPs as they are transported by the residual circulation over a vertical distance of around 50 km. Lidar observations of MSPs should also be possible in the near future with the development of novel diode-pumped alexandrite ring lasers which produce very narrow emission lines, such as used in the VAHCOLI (Vertical And Horizontal COverage by LIdar) instrument (Lübken and Höffner, 2021).

Related to the detection of AlO and sampling aerosols in the upper stratosphere and mesosphere, is the increasingly important subject of space debris. A recent study estimates that the current mass influx of space debris to the atmosphere is ~3% of the natural influx of cosmic dust, and the influx of certain metals – Al in particular – already exceeds the natural input (Schulz and Glassmeier, 2021). This study also shows that taking account of the growing requirement for satellites to be launched into low Earth orbit in order to limit their lifetimes in space, and making plausible projections of future satellite launch rates, the total mass influx from space debris is likely to approach or exceed the natural influx. Although large pieces of spacecraft entering at relatively low velocities will mostly disintegrate and ablate around 60 km and hence have a limited effect in the MLT, the effects on stratospheric ozone are a concern and should be the subject of future study.

### 7.5 Mesospheric clouds

A number of questions remain concerning the basic understanding of NLCs and their lifecycle. These concern microphysical and chemical details about the ice nucleation process, and the role of various potential condensation nuclei. However, as already indicated in Section 6, rather than being a research subject in themselves, the major future value of NLCs will most likely be to use them as an observational tool for studying other scientific questions: in particular, the use of NLCs (and PMSEs) as tracers for dynamics and variability on various spatial and temporal scales. As part of this, it will be crucial to consolidate and extend long observational time series from lidar, radar and space-borne missions. A challenge that remains is to make data series from various observational techniques and geometries compatible (Bailey et al., 2015; Broman et al., 2019).

### 8 Conclusions

This Opinion article demonstrates that the past two decades have seen major advances across a broad range of topics relating to chemistry in the MLT. Significant progress has been achieved through the "classical" combination of observations from multiple platforms, laboratory measurements and theoretical calculations of underpinning physico-chemical parameters, together with global models of increasing resolution and complexity. At the same time, this work has raised many new questions which, when combined with all the important research currently underway, promises a very exciting decade ahead. As *Atmospheric Chemistry and Physics* celebrates its 20[th] anniversary, it is worth noting that 14% of the papers cited in this article on the MLT were published in *ACP*, which is a significant achievement for a relatively young journal whose remit covers the entire Earth's atmosphere.

### Acronym List

ACE-FTS – Atmospheric Chemistry Experiment Fourier Transform Spectrometer (satellite instrument)

AIM – Aeronomy of Ice in the Mesosphere (satellite mission)

ALTIUS – Atmospheric Limb Tracker for Investigation of the Upcoming Stratosphere (satellite mission)

AWE – Atmospheric Waves Experiment (space-based imaging spectrometer)

CABMOD – Chemical Ablation MODel

CME – Coronal Mass Ejection

DA – Data Assimilation

EISCAT_3D – European Incoherent SCATter (radar system upgrade to 3-dimensional imaging)

EMAC – ECHAM/MESSy Atmospheric Chemistry (ECHAM is a general circulation model from the Max Planck Institute for Meteorology, MESSY = Modular Earth Submodel System)

GLO-1 – Arizona AirGLOw experiment (spectrograph on-board the Space Shuttle)

HALOE – Halogen Occultation Experiment (satellite mission)

HAMMONIA – HAMburg MOdel of the Neutral and Ionized Atmosphere

KASIMA – KArlsruhe SImulation model of the Middle Atmosphere

JPSS-2 – Joint Polar Satellite System 2 (satellite mission)

MAGIC – Mesospheric Aerosol Genesis, Interaction and Composition (rocket-borne dust collector)

MATS – Mesosphere Airglow/Aerosol Tomography and Spectroscopy (satellite mission)

MEPED – Medium Energy Proton and Electron Detector (space-based instrument)

MIF – Meteoric Input Function

MJO – Madden Julian Oscillation

MLT – Mesosphere and Lower Thermosphere

MSP – Meteoric Smoke Particle

NLC – NoctiLucent Cloud

OMPS – Ozone Mapping and Profiler Suite (spaceborne spectrometer)

PMC – Polar Mesospheric Cloud

PMSE – Polar Mesospheric Summer Echo

PMWE – Polar Mesospheric Winter Echo

QBO – Quasi Biennial Oscillation

SABER – Sounding of the Atmosphere using Broadband Emission Radiometry (instrument on TIMED)

SBUV – Solar Backscatter Ultraviolet Radiometer (satellite sensor)

SCIAMACHY – SCanning Imaging Absorption spectroMeter for Atmospheric CartograpHY (instrument on Envisat satellite)

SCISAT – SCIence SATellite (for Earth observation)

Sfu – solar flux unit

SNOE – Student Nitric Oxide Explorer (satellite)

SOFIE – Solar Occulation for Ice Experiment (satellite spectrometer)

SPE – Solar Proton Event

TIMED – Thermosphere Ionosphere Mesosphere Energetics Dynamics (satellite mission)

VAHCOLI – Vertical And Horizontal COverage by Lidar (ground-based lidar instrument)

WACCM – Whole Atmosphere Community Climate Model

ZoDY – Zodiacal Dust Cloud (model)

## Code/Data availability

Not applicable.

## Author contributions

All authors contributed to the writing, reviewing and editing of the article.

## Competing interests

The authors declare that they have no conflict of interest.

**Acknowledgments**

JMCP and DRM acknowledge funding from the UK Natural Environment Research Council (grant number NE/T006749/1). KSK acknowledges support from the US National Science Foundation (NSF grant AGS-2009960) and the National Aeronautics and Space Administration (NASA grant 80NSSC20K0915). CvS acknowledges financial support from the Deutsche Forschungsgemeinschaft (DFG Research Unit VolImpact, FOR 2820, grant no. 398006378).

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
