# Peer review of "Opinion: Recent Developments and Future Directions in Studying the Mesosphere and Lower Thermosphere"

_EGUsphere, 2023_

## Author Comment (AC1)

EGUSPHERE-2023-680 | **Opinion: Recent Developments and Future Directions in Studying the Chemistry of the Mesosphere and Lower Thermosphere**

John M. C. Plane, Jörg Gumbel, Konstantinos S. Kalogerakis, Daniel R. Marsh, and Christian von Savigny

**Response to the Referees**

We thank both referees for a careful reading of the manuscript and for making a number of very helpful comments for improving it. The referees' comments are italicized, and our response is in normal typeface. In the revised manuscript, the changes are shown in blue typescript.

**Referee #1**

*This paper summarizes developments in observing and understanding the composition and chemical reactions in the MLT over recent decades. There is little space given to the basic composition and dynamics of the MLT; instead, the paper assumes a working familiarity with those and focusses almost completely on recent or still unsolved aspects. The topic is broad and many details are mentioned only briefly but the authors provide an extensive list of references. The paper is very clearly written. For the most part it maintains a balance in the topics covered without undo emphasis on any particular aspect of the development.*

**Author response:** we are pleased that this referee considers that we have achieved a reasonable balance between topics. We have indeed assumed that the reader will have a working familiarity with the chemistry and physics of the MLT. Nevertheless, to take account of this and referee #2's comment about the "very short" introduction, we have now added Figure 1 which illustrates the location of the MLT and the processes affecting the chemistry and composition that are discussed in the paper. This figure is accompanied by a fairly detailed caption to introduce these topics. We have also added references to some recent review articles.

*Major comments*

*Many of the works cited in Section 3 include explanations of the dynamically coupled or dynamically driven processes that lead to various outcomes, e.g. responses to solar variability, MJO, QBO, interhemispheric coupling, volcanic eruptions, trends. These explanations are repeated in the paper without judgement. In some cases, the statistics are poor because the temporal records are short (especially the solar cycle and volcanic eruption time series), the signals are weak and irregular, and the gravity waves that are involved in many interactions are poorly observed. It is appropriate to pass along the contents of other studies in a review such as this one. In the interest of caution, a more forceful recognition that verification of many of these mechanisms awaits additional data would be a good idea. As written, the accumulation of these reports gives the impression that the variability associated with external forcing and dynamical coupling on large spatial or temporal scales is better accepted and understood than is the case.*

**Author response:** we appreciate the referee's point here and agree that some statements could be more circumscribed. We have therefore softened some statements – where appropriate – following the referee's suggestion. In addition, we added some general comments to section 3 to point out that for some of the discussed effects the underlying processes are not well understood. It is worth reiterating that our focus here is on chemistry, and so recent findings on dynamics or forcings are discussed to the extent that these are important for our understanding of MLT composition and chemistry; however, it is beyond the scope of this paper to provide an in-depth discussion of these findings.

*Minor comments*

*The section on future directions describes several measurement systems or global modeling advances that are still in planning stages or, even for those already collecting data, for which publications are not yet available. In order for readers to follow the progress in the months or years following initial publication, it would be useful to have web links for the individual programs.*

**Author response:** this is a good suggestion, and we have added links to appropriate websites for ongoing and upcoming missions where citable papers are not yet available:

AWE: https://blogs.nasa.gov/awe

ALTIUS: https://www.esa.int/Applications/Observing_the_Earth/Altius

JPSS: https://www.nesdis.noaa.gov/about/our-offices/joint-polar-satellite-system-jpss-program-office

EISCAT_3D: https://www.eiscat.se/

Meridian Space Weather Monitoring Project: https://data.meridianproject.ac.cn/

Midlatitude Allsky-imaging Network for GeoSpace Observations (MANGO): https://www.mangonetwork.org

The following references for high-resolution modelling have been added to the text: Chang et al., 2020; Hohenegger et al., 2023; Liu et al., 2014; Schwantes et al., 2022; Herrington et al., 2022. The following references have been added for upper atmospheric forecasting: Richter et al., 2022; McCormack et al., 2021.

*Editorial*

*line 38: "mbar", "mbar" Consider using SI units (Pascal) or at least use the same units for the two values being compared.*

**Author response:** changed to mbar, and added the pressure in Pascal.

*line 60-61: "Observations of NO and excited NO infrared emissions have been particularly useful in detecting the presence of atmospheric tides in the MLT."*

*I think you mean the impact of tides on composition. The tides themselves are best detected from temperature and wind observations.*

**Author response:** changed to "impact", as the referee suggests.

*line 135: "all the way to the MLT"*

**Author response:** changed to "all the way up to the MLT"

*line 520-521: "… the statistical significance of these trends remains limited"*

*This comes across as wishy-washy. Can you just say there is no significant trend?*

**Author response:** this statement is now extended to "Trends in the cloud brightness and cloud ice water content have been deduced from satellite observations (e.g., DeLand and Thomas (2015)), but

time series are affected by both solar variability and changes in stratospheric ozone, so the statistical significance of long-term trends remains limited."

**Referee #2**

*In this paper, an overview is provided in recent developments and future directions in the science of the Mesosphere and Lower Thermosphere (MLT) region. The MLT region is a highly interesting transition region between the lower and middle atmosphere, and the near-Earth space of the thermosphere-ionosphere-magnetosphere. As it is not directly accessibly by measurement platforms apart from by sounding rocket, observations of the MLT region are difficult, and many aspects of MLT processes are still not well understood. However, a lot of progress has been made in the last two decades due to advances in observation techniques and the large number of satellite observations starting 2002, and the aim of the paper is to focus on these recent advances in a selection of topics. The paper provides a very short introduction defining the MLT region and its most important properties, and then discusses observations of the MLT region from different platforms followed by a wide area of different topics: the variability of the MLT regarding energetics, wave-driven dynamics, solar forcing, and long-term trends, airglow emissions and chemistry, the cosmic dust influx and the resulting metal chemistry and cosmic dust formation, and noctilucent clouds. It ends with a discussion of future directions. The paper thus provides an interesting overview of a highly relevant, fast developing field which I quite enjoyed reading.*

*As this is an opinion paper, the selection of topics discussed is by nature somewhat subjective, depending on the choices of the authors. I have summarized a few more references and further points below ("Topical"), however would like to emphasize that these are suggestions only. A few points in my opinion should be clarified before final publication, see "Major issues", and a few minor and technical issues are listed at the end as well.*

*Major issues:*

*The title and abstract are not very clear in the sense that they do not describe the content of the paper very well. I) the paper clearly deals with much more than just the chemistry of the MLT, covering topics like wave coupling, microphysics, and energetics, but actually not covering what I would call the photochemistry of the MLT driven by photolysis, photoionization and particle impact ionization, in great detail. You could just change the title and description in the abstract as well as in two more places listed in "Minor". II) It is stated on the one hand that the paper reviews important advances over the past two decades, but with a focus on work during the past 10 years. This alleged focus is actually not clear in most of the subsections, which mainly discuss work done since 2000/2002, so in the last two decades.*

**Author response:** after some discussion between the authors, we would prefer to leave the title as it is in order to emphasise that our focus is on chemistry, but we have added the following statement in the Abstract: "Although the emphasis here is on chemistry, we also discuss recent findings on atmospheric dynamics and forcings, to the extent that these are important for understanding MLT composition and chemistry."

It is Section 3 that contains most of the "physics" discussion, but this section provides a background on atmospheric variability and dynamics that is necessary to understand the composition and the chemical processes in the MLT. There are many examples in the MLT of local conditions that are dynamically controlled, chemically controlled, or controlled by a combination of both. A paper about the composition and chemistry of the MLT is thus not feasible without a description of relevant processes governing the dynamics and variability.

We have removed the statement about focusing on work in the past 10 years, in order to be consistent with opening sentence of the Abstract: "... with a review of important advances in the science of the Mesosphere and Lower Thermosphere (MLT) region of the atmosphere that have occurred over the past two decades ...".

*In the introduction, a very short definition and overview is given about what is the mesosphere, thermosphere, and MLT region. As the topic of the publication is on recent developments in MLT science and work during the past two decades, it would have been helpful for the non-expert reader to provide also a short summary of the state of the art 10 years ago as a starting point. Some of the following chapters have a short summary like this at the beginning (e.g., chapter 3.1 which deals only with research prior to 2005), but not all; maybe some starting point like this could be provided clearly at the beginning of all chapters, or in the introduction.*

**Author response:** we have added Figure 1 with a comprehensive figure caption, and some additional references to review articles within the past 20 years or so. These additions provide the reader with more background material in the Introduction. However, we think a general description of the state-of-the-art 20 years ago in this section, covering all topics, is not really feasible given the overall length of the article. Although not always explicit, we think the reader would understand that the studies that we have chosen to cite in each of the subsequent sections represent important advances over the state-of-the-art in each topic that existed 20+ years ago.

*Chapter 3.3, lines 217-220: the high-latitude MLT region is affected by particle impact ionization of precipitating protons and electrons which come from different sources. There are solar particles, mostly protons, from solar coronal mass ejections, but there are also auroral electrons -- solar wind particles accelerated in the magnetotail -- and electrons from the radiation belts and ring currents accelerated into the loss cone e.g., in geomagnetic storms. Geomagnetic storms can be initiated by solar coronal mass ejections CMEs, but also by corotating interaction regions CIRs. Reviews of this can be found, e.g., in Sinnhuber et al., Surv. Geophys., 2012; Mironova et al., Space Sci Rev., 2015; Baker et al., Space Sci Rev., 2018; Sinnhuber and Funke, in: The dynamic loss of Earth's radiation belts, Elsevier, 2020. Precipitation of solar energetic particles in solar proton events as mentioned here can have a spectacularly large impact on the chemical composition of the middle atmosphere, but mostly in the stratosphere and lower mesosphere; in the MLT region, geomagnetic storms and auroral substorms arguably have a larger impact on, e.g., the distributions of NO, upper mesosphere OH, auroral airglow, and temperature.*

**Author response:** we agree that the text implies that only CMEs initiate particle precipitation, when it should be describing geomagnetic storms in general. The text at the start of Section 3.4 has been revised to focus on the impact of these particles on the MLT, rather than what initiates their precipitation in geospace.

*Chapter 3.4: Solar coronal mass ejections and energetic particle precipitation have already been introduced, and their effects discussed to some extent, in Section 3.3, see my above comment. However, both here and in Chapter 3.3, the different sources of energetic particle precipitation are mixed up. This should be clarified; see my summary of sources in the comment above. Normally, "space weather" refers to the disturbances of the magnetosphere-ionosphere-thermosphere by high-speed solar wind, which could come from solar coronal holes, corotating interaction regions, or solar coronal mass ejections; some but by far not all solar coronal mass ejections come with high fluxes of high-energy protons > 10 MeV, and initiate solar proton events. You appear to refer to solar proton events in line 217 and lines 252-257, but than reference publications which deal with geomagnetic activity impacts by geomagnetic storms or auroral substorms without clarifying that these are*

*different types of events with different particle sources, particle energies, and temporal evolution: e.g., Fytterer et al 2015; Hendrickx et al., 2015 in lines 221-222; and Anderssen et al., 2014; Smith-Johnsen et al., 2018 in lines 258-259 deal with geomagnetic activity, but not with solar proton event; and in lines 259-262 MEE is discussed, which are likely related mainly to geomagnetic storms.*

**Author response:** the reference to "space weather" has been replaced with the more focussed "energetic particle impacts".

*Topical:*

*Lines 58-59: there has also been interesting work on O2 airglow emissions. E.g., the derivation of the O2 emissivity from SCIAMACHY observations (Zarboo et al., AMT, 2018) has enabled a more precise observation of CO2 from space (Sun et al., GRL, 2018; Bertaux et al., AMT, 2020). SCIAMACHY O2 airglow observations have also been used to simultaneously derive O2 emissivity and MLT temperatures (Sun et al., AMT, 2022).*

**Author response:** the studies mentioned by the referee represent interesting recent work related to $O_2$ airglow emissions and their application to remote sensing of greenhouse gases. However, GHG-remote sensing is outside the scope of our paper, and so we have not modified the manuscript to include these.

*Lines 60-63: NO number density is a clear tracer of atmospheric ionization in the upper mesosphere and lower thermosphere; the abundance of NO at low and midlatitudes above 80 km altitude clearly demonstrates the role of EUV photoionization, while at high latitudes, NO corresponds to geomagnetic forcing by electron precipitation in auroral substorms and geomagnetic storms (e.g., Marsh et al.,JGR, 2004). Observations of NO density in the upper mesosphere and lower thermosphere have been carried out since 2002, e.g., by MIPAS/ENVISAT in the MA/UA modes (Bermejo-Pantaleon et al., JGR, 2011), by SCIAMACHY/ENVISAT (Bender et al., AMT, 2015; Bender et al., AMT, 2017), by ODIN/SMR (Sheese et al., JGR, 2013), ACE/FTS (Boon et al., 2005, 2013), and SNOE/AIM (Gordley et al., 2009; Hervig et al., 2019), and have been widely used to study particle impact ionization in the MLT region, e.g., showing a clear impact of geomagnetic forcing well into the mesosphere (e.g. Kirkwood et al., Ann. Geo., 2015; Hendrickx et al., JGR, 2015; Sinnhuber et al., JGR, 2016; Hendrickx et al., GRL, 2017; Smith-Johnsen et al., 2017; Kiviranta et al., ACP, 2018; Sinnhuber et al., JGR, 2022). A similar response to geomagnetic forcing has been found for OH based on observations from MLS/AURA in the high-latitude mesosphere (Verronen et al., 2011; Andersson et al., 2012; Fytterer et al., 2015). A response of mesospheric NO to geomagnetic storms and auroral substorms has also been observed by a ground-based microwave radiometer (Newnham et al., GRL, 2011; Newnham et al., JGR, 2015) (Section 2.3).*

**Author response:** indeed, in the original manuscript we did not properly cover measurements of NO from various platforms and their importance for quantifying and tracing space weather effects in the MLT. Following the referee's suggestion, we have now added corresponding text both in Section 2.1 (Satellite Observations) and Section 3.4 (Energetic Particle Impacts).

*Lines 67—74: you could add the observations of MIPAS/ENVISAT here, which also observes CO2 as well as CO, NO and T; though the full period of 10 years (2002-2012) of observations is only available for the nominal mode scanning up to 68 km, while the UA/MA modes scanning up to 170 km only started in 2005.*

**Author response:** a reference to MIPAS does not really fit into this paragraph, which is concerned with MLT observations over longer time periods than MIPAS can provide. MIPAS has contributed much to MLT research and we have added a reference earlier in this section.

*Section 2.1: radio occultation observations could be mentioned here as well, which provide global observations of electron density above ~80 km for the first time, e.g., from satellite instruments like FORMOSAT-3/COSMIC-1 (Wu et al., Remote Sensing, 2022), or a detailed view of sporadic E layers (e.g., Arras et al., Earth., Planets and Space, 2022).*

**Author response:** a sentence on sporadic *E* layer observations is now added at the end of Section 2.1

*Line 201: arguably at least as spectacular manifestations of wave coupling in the MLT are (I) Elevated Stratopause Events (ESEs), which are characterized by the re-formation of the stratopause at mesopause altitudes followed by the formation of a very strong and stable polar vortex with strong downwelling, after some SSWs (Manney et al., GRL, 2009; Orsolini et al., JGR, 2010; Siskind et al., JGR, 2010; Siskind et al., GRL, 2010; Thurairajah et al., JGR, 2010; Chandran et al., GRL, 2011; Ren et al., JGR, 2011; Limpasuvan et al., JASTP, 2012); (II) the fact that the impact of SSWs is observed even in the ionosphere (e.g., Chau et al., JGR, 2010; Goncharenko et al., GRL, 2010; Chau et al., Space Sci Rev, 2012; Goncharenko et al., GRL, 2021; Goncharenko et al., Frontiers, 2022)*

**Author response:** we prefer not to include these suggestions made by the referee. This is because the focus of the paper is on MLT chemistry, and so we want to restrict the discussion to how dynamics impacts chemistry. However, we have changed "The most spectacular manifestation" to the more correct "One of the most spectacular manifestations".

*Lines 261-265: there are a number of observations and model studies showing a temperature response to geomagnetic activity, mostly related to geomagnetic storms, but also to SPEs (Tyssoy et al., JGR, 2010; Sun et al., Universe, 2022; Wang et al., JGR, 2021; Zou et al, Astrophys. J., 2020; Li et al, GRL, 2018; Liu et al., Atmosphere, 2018).*

**Author response:** we have now added a sentence in Section 3.4 describing the temperature response to geomagnetic activity.

Minor:

*Line 40: There is also a very steep temperature gradient from the mesopause into the lower thermosphere, with changes of several hundreds of K between 80 and 120 km.*

**Author response:** we agree (and this is in fact stated earlier in Section 1), but the temperature does not approach the temperature of a molten silicate particle.

*Line 85: small-scale "vertical" structures*

**Author response:** the same applies to small-scale horizontal structures, so we prefer not to differentiate.

*Line 158: photodissociation and dissociative ionization in the EUV of O2 and N2*

**Author response:** added "and indirectly by photoionization"

*Line 174-175: The gw-driven meridional overturning circulation controls the dynamics of the upper mesosphere; however, there is a turn-around of the zonal winds (e.g., Smith et al., Surv. Geophys., 2012) and meridional winds (e.g., Wang et al.,JGR, 2022, Figure 1) around the mesopause which should separate the large-scale circulation patterns in the upper mesosphere from the lower*

*thermosphere. Large-scale transport of tracers from the lower thermosphere to the upper mesosphere appears unlikely, turbulent mixing appears more likely.*

**Author response:** we have added the following sentence which includes the Smith et al. (2012) reference suggested by the referee: "The same circulation pattern brings NO and CO from the thermosphere into the wintertime polar mesosphere and $H_2O$ and $CO_2$ upwards in the opposite pole (Garcia et al., 2014; Lossow et al., 2009; Smith, 2012)."

*Chapter 3.3, lines 212-216: the highly variable solar electromagnetic radiation in the EUV and soft x-ray range could be mentioned here as well, which is the source of the daytime ionospheric E-layer, and ionizes the daytime atmosphere down to around 80-90 km altitude.*

**Author response:** we now explicitly mention the importance of the solar cycles on vacuum and extreme ultraviolet (VUV, EUV) and soft x-rays.

*Line 222: observations of the 27-day signatures in mesospheric NO are also reported in Sinnhuber et al., JGR, 2016 and shown from model experiments in Fytterer et al., JGR, 2016. It should be noted that the 27-day signature found in NO and OH at high latitudes is likely mainly related to the 27-day signature in the solar wind, not in the electromagnetic spectrum. Insofar it might be more fitting to discuss these in the next section "Space weather impacts".*

**Author response:** we added the Sinnhuber et al. (2016) reference here, and agree that some of the solar 27-day effects are caused by variations in EM radiation, as well as by variations in the particle fluxes. This is now explicitly mentioned, but we decided to keep all the 27-day effects in this section.

*Line 231-233:.... and EUV photoionization above 90 km*

**Author response:** we have added this phrase.

*Line 261: and Sinnhuber et al., JGR, 2022 (companion paper to Tyssoy et al., 2022)*

**Author response:** we have added this reference.

*Lines 381-383: … and sporadic E-layers (in Section 5.2)*

**Author response:** done.

*Line 436-437: narrow layers of high concentrations of Fe+ and Mg+ ions and electrons*

**Author response:** done.

*Line 549: into the composition, energetics, and dynamics of the MLT, as there is discussion of e.g., gravity waves, NLCs and Airglow as well.*

**Author response:** we have added "energetics and dynamics" in parenthesis, to emphasise that these areas are not our primary focus. Section 7 already mentions future efforts concerning composition, energetics (thermal structure) and dynamics (transport processes and waves), because energetics and dynamics are relevant for the chemistry of the MLT.

*Lines 565-571: you could mention here also ESAs Earth Explorer 11 candidate mission The Changing-Atmosphere IR Tomography Explorer CAIRT (cairt.eu), an IR imager capable to simultaneously observe temperature and a large number of trace gases from the UTLS to the lower thermosphere with high spatial resolution, as a long-term perspective.*

**Author response:** information and a link about CAIRT as a possible future mission have been added.

*Lines 582-583: atomic oxygen at 63 mym has already been observed in the lower thermosphere from the CRISTA instrument on two space shuttle flights in 1994 and 1997 (Grossmann et al., GR,L 2000). Good to see that this method may be used again in the near future.*

**Author response:** this part of the paper concerns plans for future observations and we therefore list the references to Richter et al. and Yee et al. as most relevant. These papers in turn refer back to the original work by Grossmann et al. In fact, sounding rocket observations of 63 microns by Grossmann et al. date even further back than the measurements with CRISTA.

*Lines 595-601: There is also still the Network for the Detection of the Network for the Detection of Atmospheric Composition Change NDACC with some of the observations (e.g., by the microwave radiometers) reaching into the mesosphere (ndacc.larc.nasa.gov), and the Network for the Detection of Mesospheric Change NDMC (ndmc.dlr.de), which targets the mesopause region.*

*Line 723-724: see my comments above: the topic is much broader than chemistry, as it includes wave coupling, airglow, cosmic dust and noctilucent clouds.*

**Author response:** see the earlier responses – in this article we focus on chemistry, while including discussion of dynamics and forcings when these impact on chemistry and composition.

*Technical:*

*The paper would read better with using less brackets (). At least in the abstract (line 14-15, 16, 17-18), they should not be used.*

**Author response:** done

*Line 50: the acronym list is "at the end of the paper" – below suggests on this or the next page.*

**Author response:** done

*Line 54: that revealed "that" photochemistry dominates …*

**Author response:** done

*Line 106: or "a" combination of both*

**Author response:** done

*Line 308: can you clarify what "high v" means? Instead of using this, you could write "v>= 5(?)" here and in the following (e.., line 312, line 347, … ). That would be more precise.*

**Author response:** we have replaced OH(high v) by OH(high v $\geq$ 5).

*Line 342: Panka et al. (2017) appeared in …*

**Author response:** replaced "appeared" by "published".

Lines 340 and 341: we replaced "($n_3$)" by "($\nu_3$)".

---

## Author Response (AR2)

EGUSPHERE-2023-680 | **Opinion: Recent Developments and Future Directions in Studying the Chemistry of the Mesosphere and Lower Thermosphere**

John M. C. Plane, Jörg Gumbel, Konstantinos S. Kalogerakis, Daniel R. Marsh, and Christian von Savigny

**Response to the Editor's comments**

We thank the editor for a very careful reading of the manuscript and for making a number of helpful suggestions for improving it. The editor's comments are italicized, and our response is in normal typeface. In the revised manuscript with marked changes, the changes are shown using track changes.

*I have some comments of my own. See below. Please consider them and make revisions as you think appropriate. You do not need to provide full responses (but of course responses are welcome and might be interesting).*

*One referee suggested modifying the title and you argued against that. I guess that you and your co-authors are a bit worried that mesospheric experts who are not chemists will criticise this because it doesn't cover their speciality in sufficient detail. My own comment having read the paper is that there is plenty in it that is interesting to me, not a mesopheric expert, and it would be a pity if the paper was ignored by a broader readership because of the 'Chemistry of' in the title. The 'Opinion' in the title is a useful indication that this not claiming to be complete and authoritative. I would be in favour of dropping 'Chemistry of the'. The abstract could say 'The emphasis here is on chemistry, but we also discuss ...'. But it is your call.*

**Author response:** this is a very fair point. We have therefore removed "Chemistry" from the title, and changed the Abstract as suggested.

*Detailed comments:*

*40: 'pressures falling from 0.1 mbar to below 1 mubar, at which point molecular diffusion becomes more important than eddy diffusion in transporting constituent species' -- I think that you are saying that where does molecular diffusion become more important at about 1 mubar? But your previous characterisation of the MLT as up to 120km surely implies a much smaller pressure at the upper bound of that region?*

**Author response:** this sentence now reads "The MLT is therefore subject to extremes: pressures falling from 0.1 mbar to below 0.1 $\mu$bar (above the turbopause at a pressure around ~0.5 $\mu$bar, molecular diffusion becomes more important than eddy diffusion in transporting constituent species); and temperatures ranging from below 100 K (the coldest part of the planet) to over 2500 K in ablating cosmic dust particles."

*Figure 1: Good that it is included. Some parts are a bit mysterious -- e.g. I suppose that the arrows above H2O, CH4 etc imply upward transport? There are many arrows on the diagram representing many different processes. I recommend that, having taken the trouble to provide the Figure, you provide a bit more information in the caption.*

**Author response:** this is a good suggestion. The processes described by the different coloured

arrows are now identified individually in the figure caption.

*65: Why 'relatively' too dense to allow in situ measurement?*
**Author response:** "relatively" is now removed, and the sentence reads: "The atmospheric density in the MLT is too high for *in situ* measurements by satellites because of the resulting aerodynamic drag, and so satellite instruments rely on remote sensing …".

*71: 'and by the vertical transport of atomic O' would be clearer.*
**Author response:** done.

*79: 'impact of atmospheric tides in the MLT' -- you made a change in response to a referee comment. Their point was that it was in detecting the impact on composition of the tides that the emission observations had been useful. 'Impact of atmospheric tides on the MLT composition' would be closer to what the referee was recommending.*
**Author response:** done.

*82: 'SNOE' -- defined only in acronym list -- giving definition when first used, as you have done for most acronyms, would be helpful.*
**Author response:** done.

*93: 'contraction' -- also 'shrinkage' used elsewhere. Personally I don't like this kind of term because I think it can be confusing in the wider context. ('The mesosphere is shrinking!') Of course what you mean is that the physical criteria used to define the boundaries of the MLT, or boundaries within the MLT, do not correspond to fixed altitudes and changes in the temperature structure of the atmosphere therefore imply that these boundaries may change. A one-sentence explanation of this might be helpful.*
**Author response:** "Contraction" is a commonly used term in MLT science (e.g. in the title of the Mlynczak et al. (2022) paper we cite). We now define the term: "…, where contraction is defined as a decrease in the vertical distance between the same pressure levels". "Shrinkage" has been changed to "contraction" throughout the paper for consistency.

*98: Is this a definition of 'sporadic E layers'?*
**Author response:** the definition is now better defined: "Sporadic *E* layers are ionospheric irregularities that consist of high concentrations of metallic ions and electrons in narrow layers"

*138: 'A new metal' -- not really new, but newly observed in the mesosphere.*
**Author response:** agreed. This is now rephrased: "The metal Ni  has been observed for the first time…"

*149: 'metals often occur in pronounced layers' -- are these 'sporadic E layers'?*
**Author response:** no, these are neutral metal atom layers. This is now made clearer: "… these measurements have revealed that the neutral metal atoms often occur in pronounced layers, …"

*161: 'PMSE' -- give definition here as well as in acronym list.*
**Author response:** done.

*183: 'energy of a photon absorbed in the thermosphere warms the mesosphere, thereby contributing*

*to the transport of energy via thermal diffusion' -- but you seem to imply in the previous sentence that the downward diffusion of atomic oxygen is what transports the energy downward -- so isn't this 'via downward diffusion of atomic oxygen' rather than 'via thermal diffusion'?*

**Author response:** we agree this is somewhat confusing. The sentence is now shortened to: "Thus, an odd oxygen (O or $O_3$) is converted back to $O_2$ and the energy of a photon absorbed in the thermosphere warms the mesosphere."

*194: 'The same circulation pattern ...' -- 'brings NO and CO downwards from the thermosphere' would be clearer -- but it was also be clearer to re-order the sentence, since you are in the previous sentence talking about the summer pole.*

**Author response:** agreed. Now changed to "The same circulation pattern brings $H_2O$ and $CO_2$ upwards over the summer pole, and NO and CO downwards from the thermosphere into the wintertime polar mesosphere at the opposite pole ...".

*212: 'polar summer mesopause' would be clearer.*

**Author response:** done

*214: 'NLC/PMCs' -- again better to give definition of PMC here. Is 'PMC' simply an alternative term for NLC?*

**Author response:** PMCs are now defined at line 138 (in the revised manuscript).

*217: 'observational data' rather than 'experimental data'? (This wasn't a lab experiment.)*

**Author response:** done.

*231: 'still eastward' a bit odd. 'Continuing eastward'?*

**Author response:** done.

*239: 'leading to an average period of close to 27 days' not needed?*

**Author response:** this sentence has been made clearer: "The 27-day cycle is caused by the differential rotation of the Sun, with a period which is slightly variable but averages close to 27 days."

*280: '3.4 Energetic particle impacts' -- 'impact' is a word used many times in the article to mean 'effect' and we all have our favourite words. But here it gets genuinely confusing.*
*'Global observations of NO have been widely used to study particle impact ionization in the MLT region, e.g., showing a clear impact of particle forcing well into the mesosphere'*
*A reader might wonder whether the title of the section should be 'The impact of energetic particle impacts'?*

**Author response:** we agree that "impact" is used ambiguously. "Impact" is now used exclusively to refer to collisional impacts (of meteoroids, energetic particles etc.), and "effects" to refer to the consequences of various processes. Section 3.4 is renamed "Effects of energetic particle precipitation".

*304: 'quasi-2-day-wave activity' -- is the 'quasi-2-day' important? i.e. is the reader supposed to interpret something particular from the use of that term?*

**Author response:** we have added the following brief statement about this phenomenon "... (the quasi-2-day-wave is a planetary wave that is particularly significant in the mesosphere during summer) ...."

*320: 'The recent study by Hoffman et al' -- clearer as 'As noted previously, the recent study by Hoffman et al'.*
**Author response:** done

*326: Re 'shrinking', see earlier comment. Are radio reflection heights thought to correspond to particular pressure levels. Or to some other physical variables?*
**Author response:** yes, it is the altitude of the pressure level that decreases when shrinking/contraction occurs (see earlier response).

*330: 'A long-term cooling trend in the middle mesosphere appears to be well established at a rate of about 3 K decade-1 around 70 km, with perhaps a weak negative trend in the mesopause region (Beig, 2011)' -- I was a bit confused by the final part of the sentence. Is the point that the trend is clear in the middle mesosphere but not clear at the mesopause?*
**Author response:** this sentence has been rephrased to make it clearer: "A long-term cooling trend in the middle mesosphere appears to be well established at a rate of about 3 K decade$^{-1}$ around 70 km; the trend in the mesopause region is much smaller, though possibly also negative ..."

*522: 'The strong updraft connected to this circulation ...' -- personally I don't like the use of 'updraft' to describe this kind of large circulation because 'updraft' and 'downdraft' are terms that are often used in convection and I believe that the phenomenon occurring in the MLT is very different to convection. 'Strong upwelling' would be better.*
**Author response:** done.

*562: 'in the dry middle atmosphere' -- omit 'dry'? A reader might think that this refers to some part of the middle atmosphere - the 'dry' part rather than the 'wet' part.*
**Author response:** done.

*575: I've expressed reservations about use of 'shrinking' -- but actually here it seems OK because some explanation is given of what 'shrinking' actually is.*
**Author response:** changed to "contraction" for consistency with earlier discussion.

*662: 'courser-scale' > 'coarser-scale'*
**Author response:** done

*664: 'The MLT has weather, just as in the troposphere, and its evolution in time is not solely determined by external forcing either from above or below. As with any chaotic system, small changes in initial conditions can lead to large changes in the final model states.' -- this is an interesting statement, but have is there any concrete evidence on this -- i.e. that processes within the MLT itself can lead to chaotic/unpredictable behaviour? The Richter et al (2022) paper doesn't seem to address this, but perhaps I have missed something.*

**Author response:** the Richter et al. (2022) reference should have appeared earlier in the paragragh, and has now been moved. Richter et al. (2022) discuss sub-seasonal forecasting, but not on a timescale of hours, so we have inserted "potentially" into the relevant sentence. A reference to substantiate the statement about chaos leading to large changes in final states has been added. This section now reads:

"Second, it will likely become common practice to run ensembles of model simulations to capture the spread of possible atmospheric states that comes from geophysical variance (Richter et al.,

2022). The MLT has weather, just as in the troposphere, and its evolution in time is not solely determined by external forcing either from above or below. As with any chaotic system, small changes in initial conditions can lead to large changes in the final model states (Liu et al., 2009). Such ensembles can potentially be used to make forecasts of the MLT on timescales from hours to weeks, benefitting forecasting for space and tropospheric weather as well as sectors that rely on GPS and radio communications."